# THE FINAL ASCENT: WHEN BIGGER MODELS GENERALIZE WORSE ON NOISY-LABELED DATA

## ABSTRACT

Increasing the size of overparameterized neural networks has been shown to improve their generalization performance. However, real-world datasets often contain a significant fraction of noisy labels, which can drastically harm the performance of the models trained on them. In this work, we study how neural networks' test loss changes with model size when the training set contains noisy labels. We show that under a sufficiently large noise-to-sample size ratio, generalization error eventually increases with model size. First, we provide a theoretical analysis on random feature regression and show that this phenomenon occurs as the variance of the generalization loss experiences a second ascent under large noise-to-sample size ratio. Then, we present extensive empirical evidence confirming that our theoretical results hold for neural networks. Furthermore, we empirically observe that the adverse effect of network size is more pronounced when robust training methods are employed to learn from noisy-labeled data. Our results have important practical implications: First, larger models should be employed with extra care, particularly when trained on smaller datasets or using robust learning methods. Second, a large sample size can alleviate the effect of noisy labels and allow larger models to achieve a superior performance even under noise.

## 1 INTRODUCTION

Modern neural networks of ever-increasing size, with billions of parameters, have achieved an unprecedented success in various tasks. However, real-world training datasets are usually unlabeled and the commonly used crowd-sourcing and automatic-labeling techniques can introduce a lot of noisy labels (Krishna et al., 2016). Over-parameterized models can easily overfit the noisy labels and suffer a drastic drop in their generalization performance (Zhang et al., 2016). This phenomenon has inspired a recent body of work on dealing with high levels of noisy labels (Han et al., 2018; Zhang & Sabuncu, 2018; Jiang et al., 2018; Mirzasoleiman et al., 2020; Liu et al., 2020; Li et al., 2020). However, the effect of model size on neural networks' generalization performance on noisy data has been overlooked, and the following important question has remained unanswered: can large models trained on noisy-labeled data be trusted and safely used?

Contradictory to the classical view of bias-variance trade-off, increasing the size of overparameterized neural networks only improves generalization Neyshabur et al. (2014). To explain this, (Belkin et al., 2019) proposed the the *double descent* phenomenon, suggesting that the test error follows a U-shaped curve until the training set can be fit exactly, but then it begins to descend again and reaches its minimum in the overparameterized regime. This has further been investigated by a body of recent work (see Section 2) focusing on confirming or reproducing the double descent curve. Here, we instead show that label noise can change the above picture and introduce a *final ascent* to the monotonically decreasing loss curve in the overparameterized regime. Specifically, we analyze the generalization performance of models of increasing size, in terms of both *network width and density*, under varying sample size and levels of noisy labels. We find that,

*when noise-to-sample-size ratio is sufficiently large, increasing the width or density of the model beyond a certain point only hurts the generalization performance.*

We first provide theoretical evidence from random feature regression studied in prior work (Yang et al., 2020) and show that the test generalization loss can be decomposed into a decreasing bias,

a unimodal noise-independent variance and an increasing noise-dependent variance. The noise-dependent variance is more pronounced when the noise-to-sample size ratio is large, which leads to a second ascent in the total variance (*c.f.* Figures 1, and 2) and an ascent in the test loss. Interestingly, our analysis also demonstrates that under a large noise-to-sample size ratio, reducing model density by keeping a randomly selected fraction of weights can improve the generalization. Our analysis complements the double-descent phenomena, by providing a complete picture of the generalization curve vs. model size under various levels of noisy labels.

Through extensive experiments, we corroborate our theoretical results and show their validity for neural networks by showing that (1) sufficiently large label noise can lead to the final ascent in test loss and (2) sufficiently large sample size can eliminate the final ascent. In addition, we study the effect of model size on the performance of state-of-the-art methods for robust learning against noisy labels. We show that, perhaps surprisingly, the adverse effect of larger models can be observed even under a smaller noise-to-sample size ratio, when robust methods are employed. Finally, we take a closer look into the smoothness of the learned networks trained with noisy labels. We show that noisy labels can turn the previously suggested negative correlation between network size and model complexity into a positive one.

Our work is a step toward understanding the complicated overfitting and generalization behavior of modern machine learning models trained on large real-world data. In practice, our results also have several important implications: First, reducing width or dropping a fraction of (even randomly selected) weights can alleviate the effect of noisy labels. Second, when training large models, larger sample size can effectively counter the effect of noisy labels and even remove the final ascent. This explains why the harm of larger models on noisy-labeled data have not been observed by prior work (Arpit et al., 2017). Finally, large models should be used with extra care even on large datasets when training with robust methods, as larger width or density can hurt the performance of the robust learning algorithms.

## 2 RELATION TO PRIOR WORK

A bulk of recent work has studied the double descent phenomena by analyzing the error of linear regression (Hastie et al., 2019; Belkin et al., 2020; Derezinski et al., 2020), random feature regression (Mei & Montanari, 2019; Adlam & Pennington, 2020a; Yang et al., 2020; d'Ascoli et al., 2020) or even NTK regression Adlam & Pennington (2020b), showing these models can capture some important features of double descent. The primary focus of these works has been on the behavior of the total test error. A few of them decompose the test error into bias and variance (Mei & Montanari, 2019; Yang et al., 2020). Most relevant to our analysis are (Adlam & Pennington, 2020a; d'Ascoli et al., 2020) which decompose the variance into several sources. However, we take different asymptotic limit than theirs. Given $n$ training examples and $d$ input dimensions, we let $\frac{n}{d}$ tend to infinity instead of constant. The benefit of this limit is that we can derive closed-form expression of different terms in the test error and therefore obtain more interpretable result. Our setting is particularly comparable to that of (Yang et al., 2020) which analyzed the noiseless case, in the same asymptotic limit. Our result shows that label noise contributes a monotonically increasing term in the variance, in addition to the bias and noise-independent variance term derived in (Yang et al., 2020). Furthermore, we include model density into the analysis by adding another layer and show that density plays a different role than width. In noiseless setting, our Theorem 2 reproduces the empirical observation made by Golubeva et al. (2020) that increasing width while keeping the number of parameters fixed by reducing density improves generalization (see Appendix A.7), which separates the benefit of width from the effect of model capacity.

The role of label noise in double descent has been discussed before in either empirical Nakkiran et al. (2021) or theoretical Adlam & Pennington (2020a) studies. However, these works focus on the settings where double descent still holds and only highlight that label noise exacerbates the peak of test loss at the interpolation threshold. In contrast, we find that label noise can essentially change the monotonicity of the loss curve by adding a final ascent. Furthermore, we show that this ascent can be removed by using sufficiently large sample size.

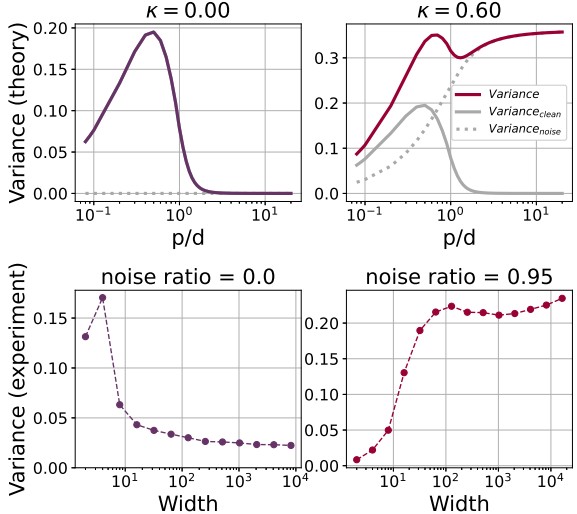

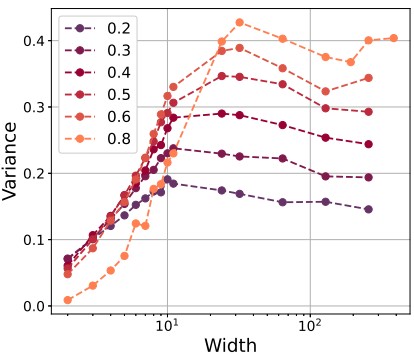

Figure 2: Variance of ResNet 34 on CIFAR-10 under different levels of label noise. For noise levels below 0.5, the variance curve is unimodal. For noise levels 0.6 and 0.8, the variance curve is unimodal until width 128 and after this point it begins to ascend again.

Figure 1: Top: the variance curve plotted based on the expressions in Theorem 1 and Appendix A.2. Bottom: variance of ReLu networks trained on MNIST.

## 3 THEORETICAL ANALYSIS OF RANDOM FEATURE REGRESSION

We first provide theoretical evidence for our main finding by analyzing the risk of a two layer neural network with a random first layer. Our result builds on (Yang et al., 2020), which studied the case where the model is trained without label noise. In particular, we extend their result to the noisy setting and show that the last ascent can be understood through the lens of bias-variance decomposition. Moreover, in Section 3.2 we show that with a simple modification to the setting, we can include model density into the analysis. It turns out that reducing the density of wide models is equivalent to using a stronger regularization in the limit, which leads to better generalization in noisy settings.

### 3.1 THE SECOND ASCENT IN VARIANCE CAUSED BY LABEL NOISE

Suppose we have a training dataset $(\boldsymbol{X}, \boldsymbol{y})$ where $\boldsymbol{X} = [\boldsymbol{x}_1, \boldsymbol{x}_2, \ldots, \boldsymbol{x}_n]$ and $\boldsymbol{y} = [y_1, y_2, \ldots, y_n]^\top$, with $\boldsymbol{x}_i \in \mathbb{R}^d$ and $y_i \in \mathbb{R}$ are feature vector and label point data point $i$. We assume each $\boldsymbol{x}_i$ is independently drawn from a Gaussian distribution $\mathcal{N}(0, \mathbf{I}_d/d)$ and labels are generated by $y_i = \boldsymbol{x}_i^\top \boldsymbol{\theta} + \epsilon_i$ where $\boldsymbol{\theta} \in \mathbb{R}^d$ with entries independently drawn from $N(0, 1)$ and $\epsilon_i \in \mathbb{R}$ is the label noise drawn from $N(0, \sigma^2)$ for each $\boldsymbol{x}_i$. The learned function is given by random feature ridge regression. Specifically, we learn a two-layer linear network where the first layer $\boldsymbol{W} \in \mathbb{R}^{p \times d}$ has each entry randomly drawn from $\mathcal{N}(0, 1/d)$ and the second layer is given by

$$\hat{\boldsymbol{\beta}} = \underset{\boldsymbol{\beta} \in \mathbb{R}^p}{\arg\min} \|(\boldsymbol{W}\boldsymbol{X})^\top \boldsymbol{\beta} - \boldsymbol{y}\|^2 + \lambda \|\boldsymbol{\beta}\|^2 = (\boldsymbol{W}\boldsymbol{X}\boldsymbol{X}^\top \boldsymbol{W}^\top + \lambda \mathbf{I})^{-1} \boldsymbol{W}\boldsymbol{X}(\boldsymbol{X}^\top \boldsymbol{\theta} + \boldsymbol{\epsilon}),$$

where $\boldsymbol{\epsilon} = [\epsilon_1, \epsilon_2, \ldots, \epsilon_n]^\top$. Given a clean test example $(\boldsymbol{x}, y)$ where $\boldsymbol{x} \sim \mathcal{N}(0, \mathbf{I}_d/d)$ and $y = \boldsymbol{x}^\top \boldsymbol{\theta}$, the prediction of the learned model is given by $f(\boldsymbol{x}) = (\boldsymbol{W}\boldsymbol{x})^\top \hat{\boldsymbol{\beta}}$. The expected risk can then be written as

$$\textbf{Risk} = \mathbb{E}_{\boldsymbol{\theta}} \mathbb{E}_{\boldsymbol{x}} \mathbb{E}_{\boldsymbol{X}, \boldsymbol{W}, \boldsymbol{\epsilon}} (f(\boldsymbol{x}) - y)^2 = \underbrace{\mathbb{E}_{\boldsymbol{\theta}} \mathbb{E}_{\boldsymbol{x}} (\mathbb{E}_{\boldsymbol{X}, \boldsymbol{W}, \boldsymbol{\epsilon}} f(\boldsymbol{x}) - y)^2}_{\textbf{Bias}^2} + \underbrace{\mathbb{E}_{\boldsymbol{\theta}} \mathbb{E}_{\boldsymbol{x}} \mathbb{V}_{\boldsymbol{X}, \boldsymbol{W}, \boldsymbol{\epsilon}} f(\boldsymbol{x})}_{\textbf{Variance}} \quad (1)$$

$$= \underbrace{\frac{1}{d} \|\mathbb{E}_{\boldsymbol{X}, \boldsymbol{W}} \boldsymbol{B} - \mathbf{I}\|_F^2}_{\textbf{Bias}^2} + \underbrace{\frac{1}{d} \mathbb{E}_{\boldsymbol{X}, \boldsymbol{W}} \|\boldsymbol{B} - \mathbb{E}_{\boldsymbol{X}, \boldsymbol{W}} \boldsymbol{B}\|_F^2}_{\textbf{Variance}_{\text{clean}}} + \underbrace{\frac{\sigma^2}{d} \mathbb{E}_{\boldsymbol{X}, \boldsymbol{W}} \|\boldsymbol{A}\|_F^2}_{\textbf{Variance}_{\text{noise}}},$$

where $\boldsymbol{A} := \boldsymbol{W}^\top (\boldsymbol{W}\boldsymbol{X}\boldsymbol{X}^\top \boldsymbol{W}^\top + \lambda \mathbf{I})^{-1} \boldsymbol{W}\boldsymbol{X}$ and $\boldsymbol{B} := \boldsymbol{A}\boldsymbol{X}^\top$. See the derivation in Appendix A.1. Here we first decompose the risk into bias and variance and then further decompose the variance into two terms where the second term $\textbf{Variance}_{\text{noise}}$ captures the role of label noise.

Our analysis is performed in the high-dimensional asymptotic limit where $n, d, p \to \infty$, with $\frac{n}{d} = \psi$, $\frac{p}{d} = \gamma$, $\frac{\sigma^2}{(n/d)} = \kappa$ held constant. To further simplify the analysis we let $\psi = \infty$. Note that here the noise-to-sample size ratio $\kappa$ is still finite. Yang et al. (2020) analyzed bias-variance decomposition in the same setting without considering label noise. Therefore we can simply use their expressions for **Bias**$^2$ and **Variance**$_{\text{clean}}$, in Appendix A.2. **Bias**$^2$ is monotonically decreasing and **Variance**$_{\text{clean}}$ is unimodal. The explicit expression for **Variance**$_{\text{noise}}$ is given below.

**Theorem 1.** *For a 2-layer linear network with $p$ hidden neurons and a random first layer, consider learning the second layer by ridge regression with parameter $\lambda$ on $n$ training examples with feature dimension $d$ and label noise with variance $\sigma$, let $\lambda = \frac{n}{d}\lambda_0$ and $\sigma^2 = \frac{n}{d}\kappa$ for some fixed $\lambda_0$ and $\kappa$. The asymptotic expression (where $n, d, p \to \infty$ with $\frac{n}{d} = \infty$ and $\frac{p}{d} = \gamma$) of* **Variance**$_{\text{noise}}$ *is given by*

$$\textbf{\textit{Variance}}_{noise} = \frac{\kappa}{2}\left(\gamma + 2\lambda_0 + 1 - \frac{\gamma^2 + (3\lambda_0 - 2)\gamma + 2\lambda_0^2 + 3\lambda_0 + 1}{\sqrt{\gamma^2 + (2\lambda_0 - 2)\gamma + \lambda_0^2 + 2\lambda_0 + 1}}\right)$$

We plot the above expression as well as the expression of **Variance**$_{\text{clean}}$ in the top row of Figure 1. We set $\lambda_0 = 0.075$. **Variance**$_{\text{noise}}$ monotonically increases and **Variance**$_{\text{clean}}$ is unimodal. Since **Variance**$_{\text{noise}}$ scales with $\kappa$, a second ascent in the variace occurs when $\kappa$ is large enough. As result, the risk, which is the sum of **Variance** and monotonically decreasing **Bias**$^2$ (Figure 3), finds its minimum at intermediate width.We show the derivation of **Variance**$_{\text{noise}}$ in Appendix A.3. Alternatively, one can also obtain this

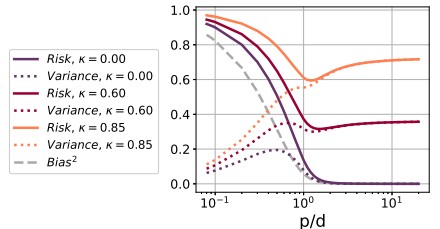

Figure 3: Risk, bias and variance given by expressions in section A.2 and Theorem 1.

result by applying Adlam & Pennington (2020a)'s Lemma 1 and Theorem 1 and taking the same limit as ours, since our **Variance**$_{\text{noise}}$ here is the sum of their $V_\epsilon$, $V_{P\epsilon}$, $V_{X\epsilon}$ and $V_{PX\epsilon}$. We note that the final ascent phenomenon has not been revealed by Adlam & Pennington (2020a) since they only studied the shape of the loss curve for the ridgeless setting where the loss only decreases in the overparameterization regime regardless of the noise.

### 3.2 Incorporating Model Density into the Analysis

When the width of the model is beyond optimal, instead of directly reducing the width, one can mask a fixed set of the parameters throughout training to have fewer trainable parameters. We call this reducing the model *density*, with *density* defined as the fraction of remaining parameters. We study the effect of density for two reasons: First, changing density gives us more fine-grained capacity control. In practice, by changing width one can only get (width $\times m$) parameters where $m$ is the number of parameters at width 1. However, changing density can yield arbitrary number of parameters $\leq$ (width $\times m$) and therefore achieves more fine-grained and architecture-independent capacity control. This is even more important for heavily engineered architectures whose width is particularly hard to reduce, e.g. InceptionResNet-v2 Szegedy et al. (2017), where we experiment with in Section 4.3. Second, As we will show below (Figure 4d), under smaller density we can always find lower test loss than the test loss achieved by the optimal density at a larger width.

In our analysis below, we consider the case where the set of trainable weights are selected randomly instead of following any weight selection strategies. This is to make sure the selected subset of weights are independent of other factors, e.g., initialization and the data, since otherwise the observed change in generalization can not be merely attributed to the number of parameters.

The setting of Theorem 1 where the first layer of the network is random and the second layer is trained can not distinguish reducing density from reducing width because the parameter of the second layer is a vector. Therefore we instead consider a three-layer linear network where the first layer is the same as in Theorem 1 whose output yields random features, the second layer is randomly masked and trained, and the last layer is also random. Formally, the function represented by the network is $f(\boldsymbol{x}) = ((\boldsymbol{V} \odot \boldsymbol{M})\boldsymbol{W}\boldsymbol{x})^\top \boldsymbol{\mu}$. The first layer parameter $\boldsymbol{W}$ and input data $\boldsymbol{X}$ are the same as in Theorem 1. $\boldsymbol{V} \in \mathbb{R}^{q \times p}$ is the second-layer parameters and $\boldsymbol{M} \in \{0, 1\}^{q \times p}$ is the mask applied to $\boldsymbol{V}$, whose entries are independently drawn from $Bernoulli(\alpha)$ where $\alpha \in [0, 1]$ is the density. $\boldsymbol{\mu} \in \mathbb{R}^q$ is the last-layer parameter whose entries are independently drawn from $\mathcal{N}(0, 1/q)$ (we set

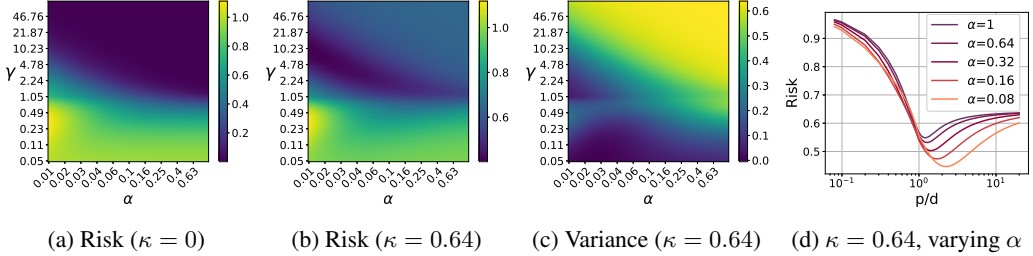

(a) Risk ($\kappa = 0$)     (b) Risk ($\kappa = 0.64$)     (c) Variance ($\kappa = 0.64$)     (d) $\kappa = 0.64$, varying $\alpha$

Figure 4: (a) Risk decreases along both axes when $\kappa = 0$. When $\kappa = 0.64$, (b) optimal risk is achieved by intermediate width and density because (c) variance increases along both axes when width is sufficiently large. (d) For $\alpha_1 < \alpha_2$, there exist models at density $\alpha_1$ whose risk is smaller than the risk achieved by optimal width at density $\alpha_2$.

the variance to $1/q$, so that $q$ does not appear in the final expressions and we can plot the risk against only $\alpha$ and $\gamma$ in a 2-D plane. We study other variants, e.g, set it to $1/d$ and let $q = p$ in Appendix A.6). We show that, the above setting, in terms of the risk and its decomposition, is equivalent to replacing the $\lambda_0$ in Theorem 1 with $\lambda_0/\alpha$. See the proof in Appendix A.4.

**Theorem 2.** *For a 3-layer linear network with p first-layer hidden neurons and both the first and last layers random, consider learning the second layer by ridge regression with random masks drawn from $Bernoulli(\alpha)$. Let $\lambda$, $n$ and $\sigma$ be the ridge regression parameter, number of training examples and noise level, respectively. Let $\lambda = \frac{n}{d}\lambda_0$ and $\sigma^2 = \frac{n}{d}\kappa$ for some fixed $\lambda_0$ and $\kappa$. The asymptotic expressions (where $n, d, p \to \infty$ with $\frac{n}{d} = \infty$ and $\frac{p}{d} = \gamma$) of **Risk**, **Bias**$^2$, **Variance**$_{clean}$ and **Variance**$_{noise}$ are given by their counterparts in Theorem 1 with $\lambda_0$ substituted with $\lambda_0/\alpha$.*

We plot the risk and variance given by Theorem 2 in Figure 4. We set $\lambda_0$ to 0.05. Plots of variance under $\kappa = 0$ (**Variance**$_{clean}$), bias, decomposition of variance are in Appendix A.5. Under zero noise, the risk monotonically decreases along both axes. In contrast, when $\kappa$ is large, both the risk and variance increase along both axes when width and density are large. In Figure 4d we further show that tuning both width and density can achieve better generalization than only tuning width. Therefore, when looking for an optimal model capacity, it is necessary to consider the effect of density.

We note that our theorem also subsumes the phenomenon empirically observed by Golubeva et al. (2020) that increasing width while keeping the number of parameters fixed improves generalization, although this is not the main focus of this paper. And sufficiently large noise can change this picture (see risk curve with fixed $\alpha\gamma$ in Appendix A.7).

## 4   EXPERIMENTS ON NEURAL NETWORKS — WHEN LARGER MODELS GENERALIZE WORSE ON NOISY-LABELED DATA

The main purpose of this section is to provide empirical evidence for our theoretical analysis. We show that the final ascent in test loss can be observed across datasets, model architectures and noise types. In Section 4.2.1, we also demonstrate that the *second ascent in variance* (analyzed in section 3.1 for linear models) occurs for neural networks, providing insights to understand the observed generalization behavior through bias-variance trade-off.

### 4.1   EXPERIMENTAL SETUP

**Noise Type.** Our theoretical example (Section 3) is a regression problem where label noise follows a Gaussian distribution. The experiments below are for classification tasks since it is the most commonly considered task for neural networks and also empirically studied for confirming the double descent phenomenon Belkin et al. (2019); Yang et al. (2020); Nakkiran et al. (2021). We consider three different noise types. On MNIST, CIFAR-10 and CIFAR-100, if not stated otherwise, the noise type is random label flipping (also called symmetric noise). In addition, we consider asymmetric noise on CIFAR-10 and CIFAR-100 where wrong labels are generated in a class-dependent way to mimic real-world noise. Images in Red Stanford Car are crawled from the web and therefore have real-world web noise.

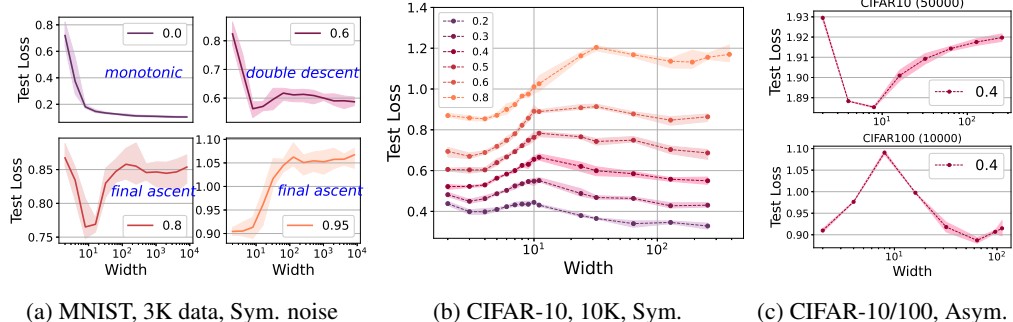

Figure 5: Test loss on (a) MNIST with 3K training examples and symmetric noise, (b) CIFAR-10 with 10K training examples and symmetric noise, (c) CIFAR-10 with 50K (all) training examples and CIFAR-100 with 10K training examples, both with asymmetric noise.

**MNIST and CIFAR-10 with symmetric noise.** We train two-layer ReLU networks on MNIST LeCun (1998) and ResNet34 (He et al., 2016) on CIFAR-10 (Krizhevsky et al., 2009). The width of the two-layer network is controlled by the number of hidden neurons. And the width of ResNet is controlled by the number of convolutional layer filters: for width $w$, there are $w, 2w, 4w, 8w$ filters in each layer of the four Residual Blocks, respectively. **Measuring bias and variance:** To empirically verify Theorem 1 which says label noise can lead to a second ascent in the variance, we measure the variance using the unbiased estimator proposed in Yang et al. (2020): we randomly split the training set for each model into $N$ subsets ($N = 20$ for MNIST and $N = 5$ for CIFAR-10), train a copy of the model for each subset and compute the variance of the network output across the subsets. In both settings we use MSE loss because the bias-variance decomposition is only well-defined for MSE (see Eq. 1). (Yang et al., 2020) also proposed an estimator for Cross Entropy loss, but it is biased and therefore may skew the results. Details of training hyperparameters are in Appendix B.

**CIFAR10/100 with asymmetic noise.** We also perform experiments on CIFAR-10 and CIFAR-100 with 40% asymmetric noise. Details of noise generalization can be found in Appendix B.1. We use Cross Entropy loss.

**Red Stanford Car.** We train InceptionResNet-v2 (Szegedy et al., 2017) on Red Stanford Cars (Jiang et al., 2020) that contains controlled real-world web noise. We consider noise levels 30% and 70%. The details of the dataset are in Appendix B.1 and the training setup is in Appendix B.2. Due to the intricate nature of the model architecture, we only vary the density of this model. For each density, we repeat the experiment 4 times and take the average of the performance.

## 4.2 LARGER WIDTH HURTS GENERALIZATION

We first train models of increasing width under various levels of label noise. Results are presented in Figure 5. Under small label noise, the test loss is either monotonically decreasing (0% noise in 5a) or double descent (60% noise in 5a, 20% and 40% noise in 5b). When the noise level is sufficiently large, there is eventually a width beyond which the test loss increases. For MNIST with 80% and 95% label noise (Figure 5a), and CIFAR-10 with 60% and 80% noise (Figure 5b), the final ascent occurs and the loss curve is a double U-shape, with the globally optimal width in the under parameterized regime, i.e., the bottom of the first U-shape. Figure 5c shows the final ascent under asymmetric noise. We note that the noise threshold where the final ascent happens can depend on both dataset and noise type. Plots of the test accuracy can be found in appendix C.1.

### 4.2.1 THE SECOND ASCENT IN VARIANCE

Our next experiment shows that the behavior shown in Figure 5 can be explained through bias-variance decomposition, as is in our theoretical example. In Section 3.1, we attributed the final ascent in test loss to the second ascent in total variance for random feature regression. Here, the same applies to neural networks trained on MNIST (Figures 1 bottom row) and CIFAR-10 (Figure 2). The shape of the variance curve is exactly as predicted by our theorem (Figure 1 top row). Plots of the bias curve can be found in appendix C.1.

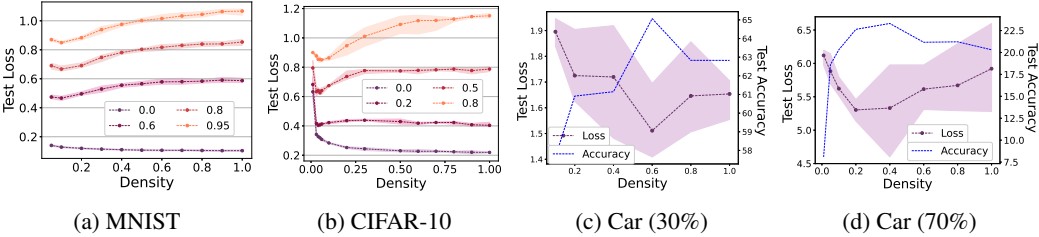

(a) MNIST          (b) CIFAR-10          (c) Car (30%)          (d) Car (70%)

Figure 6: Test performance models with varied densities. Increasing noise level changes the test loss curve from decreasing to more like a U-shaped. Therefore the optimal generalization is achieved by models at intermediate density.

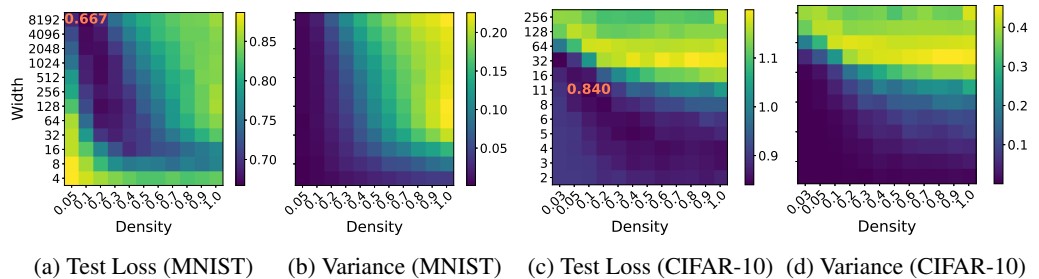

(a) Test Loss (MNIST)   (b) Variance (MNIST)   (c) Test Loss (CIFAR-10)   (d) Variance (CIFAR-10)

Figure 7: Test loss (a),(c) and variance (b),(d) of models at varied widths and densities. For MNIST the lowest loss is achieved by width 8192 and density 0.1. For CIFAR-10 the lowest loss is achieved by width 11 with density 0.1. In both settings, the variance increases with both density and width after certain point. The trend of both loss and variance is predicted by Theorem 2 (Figure 4).

### 4.3 SMALLER DENSITY HELPS GENERALIZATION

Next we vary the model density. At initialization, we randomly select a certain fraction of weights and keep them zero throughout the training. Figure 6 shows how the performance changes with density under different noise levels. Under small label noise, test loss decreases when density increases. Under larger label noise, the test loss is U-shaped. Notably, for InceptionResNet-v2 (Figure 6d), there is no straightforward way to reduce the width, however reducing density allows us to further control the capacity and achieve better generalization.

**Smaller density vs. stronger regularization.** One may notice that Theorem 2 implies reducing density is equivalent to increasing the strength of $l_2$ regularization by the inverse factor. Such equivalence may not hold for neural networks considering the non-linearity and the non-convexity. Interestingly, our experiment shows that reducing density can achieve a better generalization than increasing $l_2$ regularization. We defer the plots and discussion to Appendix C.3.

#### 4.3.1 JOINT EFFECT OF DENSITY AND WIDTH

We conduct further experiments on MNIST and CIFAR-10 with 80% label noise and vary both width and density. Comparing Figure 7 with Figures 4b and 4c confirms that empirical results align well with our theory. Figures 7b and 7d show the variance of the test loss. In both figures there is an increasing trend towards the top-right corner, though for CIFAR-10 the behavior is non-monotonic and more complicated. Figures 7a and 7c demonstrate the test loss with the annotated number in red showing which model achieves the lowest test loss. As we can see, there is an increasing trend in the loss towards the top-right corner. Notably, in both settings the lowest loss is achieved by models with small densities. This happens for two reasons: (1) reducing density is a more fine-grained capacity control; (2) as suggested by our theoretical example (Figure 4d), models of optimal width under smaller density has better generalization. Plots for the test accuracy and monotonically decreasing (along both axes) bias are in Figure 17 Appendix C.2.

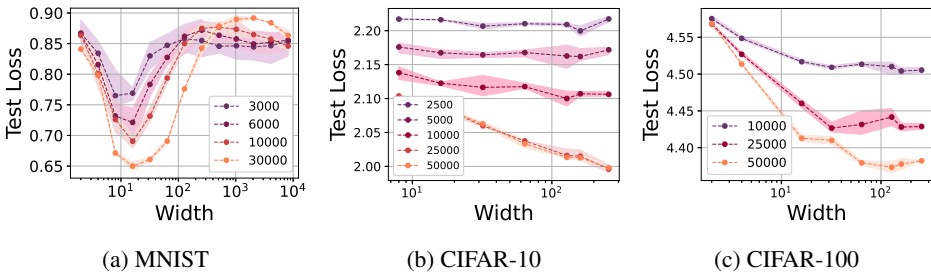

Figure 8: Legend shows number of training examples. As dataset size grows, test loss decreases and the final ascent becomes less pronounced.

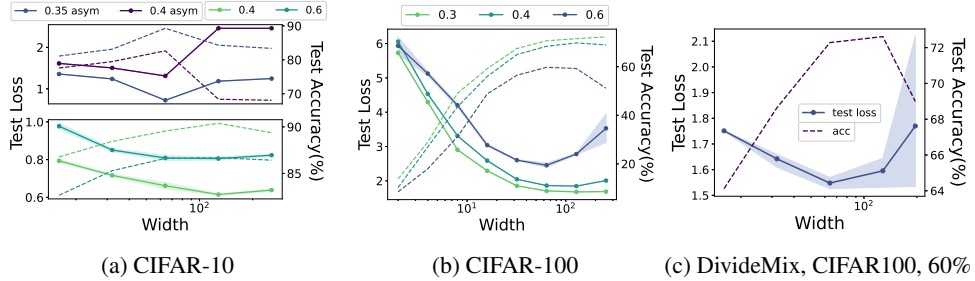

Figure 9: Test performance of models trained with ELR and DivideMix. Solid line shows test loss and dashed line shows test accuracy. The ascent in the loss can still be observed here. By comparing (a) with Figures 8b and 8c, we see that ELR even makes the phenomenon more pronounced.

## 5 EFFECT OF SAMPLE SIZE ON GENERALIZATION

In Theorem 1, the scale of **Variance**$_{noise}$ is controlled by the noise-to-sample-size ratio instead of noise itself. This implies that increasing sample size alleviates the effect of noise. In this section, we explore whether the same applies to neural networks. We fix the noise ratio and vary the number of training examples. The experiment on MNIST is in the same setting as in Section 4 and we set noise level to $80\%$. For CIFAR-10 and CIFAR-100 we use Cross Entropy loss and the experimental details can be found in B.3. We consider 60% symmetric noise. We plot the test loss curve in Figure 8. On MNIST and CIFAR-10, the final ascent is less pronounced and eventually removed as we increase the sample size. However, on CIFAR-100 the final ascent still occurs even on the full dataset (although the test accuracy shows a different pattern (see Appendix C.4). We suspect it is because CIFAR-100 has fewer examples (and thus larger fraction of wrong labels) per class, which leads the shape of the curve to be affected more drastically by noise.

**Discussion.** Nakkiran et al. (2021) empirically observed that increasing sample size shrinks the area under the double descent curve and shifts the curve to the right. This suggests the possibility that in our case the final ascent is only shifted, i.e., it still occurs but at a size larger than the largest width in Figure 8, rather than being eliminated. Although it is impossible to exclude such possibility through experiments, we note that, this possibility is not supported by our theorem that implies sufficiently large sample size should completely remove the final ascent.

## 6 WHEN ROBUST ALGORITHMS ARE APPLIED

The above experiments are all conducted with standard training procedures. In practice, robust learning methods are usually applied to achieve satisfactory test performance when the noise level is high. One may expect the final ascent to be mitigated in this case, since robust methods are specially designed to encounter the effect of label noise. However, perhaps surprisingly, we find that robust methods only make the final ascent more pronounced. We experiment with two SOTA algorithms, ELR Liu et al. (2020) and DivideMix Li et al. (2020). We train ResNet34 on full CIFAR-10 and CIFAR-100 with the same training setup as reported in the original papers (Liu et al., 2020; Li et al., 2020) (see details in Appendix B.4) and plot the test performance in Figure 9. When the models are tained on CIFAR-10/100 with ELR, the final ascent occurs under only 40% noise. In Figure

9a we also see that asymmetric noise exacerbates the phenomenon. When models are trained with DivideMix, the final ascent occurs under 60% symmetric noise on CIFAR-100 (Figure 9c) but does not occur on CIFAR-10 as is also the case with standard training procedure. Experiments for more noise levels are shown in Appendix C.5. The density experiments are also in Appendix C.5, which includes training InceptionResNet-v2 with ELR on Red Stanford Car.

# 7    A CLOSER LOOK INTO THE COMPLEXITY OF LEARNED FUNCTIONS

A hypothesis for double descent made by Belkin et al. (2019) is: the learning algorithm happens to have the right inductive bias towards the 'low complexity' function which generalizes better while fitting the training set. A larger model size, i.e., a richer function class, provides more choices for the algorithm thus letting it find a function with lower complexity. Given our finding that large noise flips the correlation between model size and generalization, it is natural to ask 'does large noise also flip the correlation between model size and the complexity of learned functions?'. In this section, we provide empirical evidence suggesting that the answer is yes. Recently, Kalimeris et al. (2019) qualitatively measured complexity of neural networks based on how much their prediction can be explained by a smaller model. This can be problematic in our case since it already

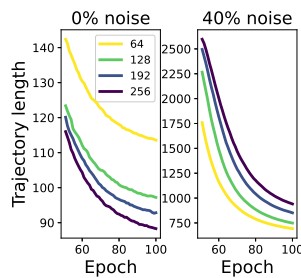

Figure 10: Trajectory Length

takes 'smaller models have lower complexity' as a premise. We instead make use of the result in (Loukas et al., 2021), which shows bias parameter of the first layer of a neural network travels longer during training when it is fitting a more complex function (a function with larger Lipschitz constant on the data). We use the same setting as in Section 6 of Loukas et al. (2021) to get tight estimates for complexity: we train a CNN with one identify layer plus 2 convolutional layers on CIFAR-10, DOG vs AIRPLANE classes. The per epoch bias trajectory length of models at different widths is plotted in Figure 10. The result shows that wider models have shorter trajectory length on clean dataset. Thus, a larger size implies lower complexity of the learned function. In contrast, with 40% label noise, larger models have longer trajectory thus learning functions of higher complexity. We refer the readers to Appendix D for more experimental results including: measuring variance of the first layer bias (which is also correlated with complexity Loukas et al. (2021)) and spectral norm (known as an upper bound for the Lipschitz constant (Szegedy et al., 2013)) and the above measures at different densities.

# 8    CONCLUSION

Our work provides new insights into the overfitting and generalization of neural networks by showing a counter example for the recently observed generalization benefits of increasing model size. We considered two ways of varying model size: changing width and randomly dropping a fraction of trainable parameters. We made two key observations: (1) when noise is sufficiently large, generalization gets worse if we increase the model size beyond some point, leading to the final ascent; (2) sufficiently large sample size can alleviate or even eliminate the final ascent. We provided theoretical evidence from random feature regression and explain such phenomenon through the lens of bias-variance decomposition. We also performed extensive experiments verifying that the above holds for neural networks. Additionally, training models with SOTA robust methods, namely ELR and DivideMix, does not mitigate the final ascent but even exacerbates it in some settings. We also provided empirical evidence which suggests the originally positive correlation between size and complexity of learned functions can be flipped by label noise. While our analysis in this paper is limited to label noise, it would be valuable to investigate whether same conclusion holds for input feature noise.

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

## A  THEORETICAL RESULTS

**Notations:** We use bold-faced letter for matrix and vectors. The training dataset is denoted by $(\boldsymbol{X}, \boldsymbol{y})$ where $\boldsymbol{X} = [\boldsymbol{x}_1, \boldsymbol{x}_2, \ldots, \boldsymbol{x}_n]$ and $\boldsymbol{y} = [y_1, y_2, \ldots, y_n]^\top$, with $\boldsymbol{x}_i \in \mathbb{R}^d$ and $y_i \in \mathbb{R}$ are feature vector and label point data point $i$. We assume each $\boldsymbol{x}_i$ is independently drawn from a Gaussian distribution $\mathcal{N}(0, \mathbf{I}_d/d)$ and labels are generated by $y_i = \boldsymbol{x}_i^\top \boldsymbol{\theta} + \epsilon_i$ where $\boldsymbol{\theta} \in \mathbb{R}^d$ with entries independently drawn from $N(0, 1)$ and $\epsilon_i \in \mathbb{R}$ is the label noise drawn from $N(0, \sigma^2)$ for each $\boldsymbol{x}_i$. And we assume each test example $(\boldsymbol{x}, y)$ is clean, i.e., $y = \boldsymbol{x}^\top \boldsymbol{\theta}$.

### A.1  BIAS-VARIANCE DECOMPOSITION OF THE MSE LOSS IN SECTION 3

Here we show the derivation of Equation 1.

$$
\begin{aligned}
\mathbf{Bias}^2 =& \mathbb{E}_{\boldsymbol{\theta}} \mathbb{E}_{\boldsymbol{x}} (\mathbb{E}_{\boldsymbol{X}, \boldsymbol{W}, \epsilon} f(\boldsymbol{x}) - y)^2 \\
=& \mathbb{E}_{\boldsymbol{\theta}} \mathbb{E}_{\boldsymbol{x}} (\mathbb{E}_{\boldsymbol{X}, \boldsymbol{W}, \epsilon} ((\boldsymbol{W}\boldsymbol{x})^\top \hat{\boldsymbol{\beta}}) - y)^2 \\
=& \mathbb{E}_{\boldsymbol{\theta}} \mathbb{E}_{\boldsymbol{x}} [\boldsymbol{x}^\top (\mathbb{E}_{\boldsymbol{X}, \boldsymbol{W}, \epsilon} (\boldsymbol{B}\boldsymbol{\theta} + \boldsymbol{A}\epsilon) - \boldsymbol{\theta})]^2 \\
=& \mathbb{E}_{\boldsymbol{\theta}} \mathbb{E}_{\boldsymbol{x}} \operatorname{Tr}[(\mathbb{E}_{\boldsymbol{X}, \boldsymbol{W}, \epsilon} (\boldsymbol{B}\boldsymbol{\theta}) - \boldsymbol{\theta})^\top \boldsymbol{x}\boldsymbol{x}^\top (\mathbb{E}_{\boldsymbol{X}, \boldsymbol{W}, \epsilon} (\boldsymbol{B}\boldsymbol{\theta}) - \boldsymbol{\theta})] \\
=& \mathbb{E}_{\boldsymbol{\theta}} \operatorname{Tr}[(\mathbb{E}_{\boldsymbol{X}, \boldsymbol{W}, \epsilon} (\boldsymbol{B}\boldsymbol{\theta}) - \boldsymbol{\theta})^\top \mathbb{E}_{\boldsymbol{x}} (\boldsymbol{x}\boldsymbol{x}^\top) (\mathbb{E}_{\boldsymbol{X}, \boldsymbol{W}, \epsilon} (\boldsymbol{B}\boldsymbol{\theta}) - \boldsymbol{\theta})] \\
=& \frac{1}{d} \mathbb{E}_{\boldsymbol{\theta}} \| \mathbb{E}_{\boldsymbol{X}, \boldsymbol{W}, \epsilon} (\boldsymbol{B}\boldsymbol{\theta}) - \boldsymbol{\theta} \|_F^2 \\
=& \frac{1}{d} \mathbb{E}_{\boldsymbol{\theta}} \operatorname{Tr}[(\mathbb{E}_{\boldsymbol{X}, \boldsymbol{W}} \boldsymbol{B} - \mathbf{I}) \boldsymbol{\theta}\boldsymbol{\theta}^\top (\mathbb{E}_{\boldsymbol{X}, \boldsymbol{W}} \boldsymbol{B} - \mathbf{I})] \\
=& \frac{1}{d} \operatorname{Tr}[(\mathbb{E}_{\boldsymbol{X}, \boldsymbol{W}} \boldsymbol{B} - \mathbf{I}) \mathbb{E}_{\boldsymbol{\theta}} (\boldsymbol{\theta}\boldsymbol{\theta}^\top) (\mathbb{E}_{\boldsymbol{X}, \boldsymbol{W}} \boldsymbol{B} - \mathbf{I})] \\
=& \frac{1}{d} \| \mathbb{E}_{\boldsymbol{X}, \boldsymbol{W}} \boldsymbol{B} - \mathbf{I} \|_F^2 \\
\mathbf{Variance} =& \mathbb{E}_{\boldsymbol{\theta}} \mathbb{E}_{\boldsymbol{x}} \mathbb{V}_{\boldsymbol{X}, \boldsymbol{W}, \epsilon} f(\boldsymbol{x}) \\
=& \mathbb{E}_{\boldsymbol{\theta}} \mathbb{E}_{\boldsymbol{x}} \mathbb{E}_{\boldsymbol{X}, \boldsymbol{W}, \epsilon} [\boldsymbol{x}^\top (\boldsymbol{B}\boldsymbol{\theta} + \boldsymbol{A}\epsilon) - \mathbb{E}_{\boldsymbol{X}, \boldsymbol{W}, \epsilon} \boldsymbol{x}^\top (\boldsymbol{B}\boldsymbol{\theta} + \boldsymbol{A}\epsilon)]^2 \\
=& \frac{1}{d} \mathbb{E}_{\boldsymbol{\theta}} \mathbb{E}_{\boldsymbol{X}, \boldsymbol{W}, \epsilon} \| (\boldsymbol{B}\boldsymbol{\theta} + \boldsymbol{A}\epsilon) - \mathbb{E}_{\boldsymbol{X}, \boldsymbol{W}} \boldsymbol{B}\boldsymbol{\theta} \|_F^2 \\
=& \frac{1}{d} \mathbb{E}_{\boldsymbol{\theta}} \mathbb{E}_{\boldsymbol{X}, \boldsymbol{W}, \epsilon} \| (\boldsymbol{B} - \mathbb{E}_{\boldsymbol{X}, \boldsymbol{W}} \boldsymbol{B}) \boldsymbol{\theta} + \boldsymbol{A}\epsilon \|_F^2 \\
=& \frac{1}{d} \mathbb{E}_{\boldsymbol{\theta}} \mathbb{E}_{\boldsymbol{X}, \boldsymbol{W}} \| (\boldsymbol{B} - \mathbb{E}_{\boldsymbol{X}, \boldsymbol{W}} \boldsymbol{B}) \boldsymbol{\theta} \|_F^2 + \frac{1}{d} \mathbb{E}_{\boldsymbol{X}, \boldsymbol{W}, \epsilon} \| \boldsymbol{A}\epsilon \|_F^2 \\
=& \underbrace{\frac{1}{d} \mathbb{E}_{\boldsymbol{X}, \boldsymbol{W}} \| \boldsymbol{B} - \mathbb{E}_{\boldsymbol{X}, \boldsymbol{W}} \boldsymbol{B} \|_F^2}_{\textbf{Variance}_{\text{clean}}} + \underbrace{\frac{\sigma^2}{d} \mathbb{E}_{\boldsymbol{X}, \boldsymbol{W}} \| \boldsymbol{A} \|_F^2}_{\textbf{Variance}_{\text{noise}}}
\end{aligned}
$$

### A.2  EXPRESSIONS OF BIAS AND VARIANCE_CLEAN

Yang et al. (2020) analyzed the bias-variance decomposition without considering label noise. Hence the variance in their analysis corresponds to **Variance**$_{\text{clean}}$ in ours. We show their expressions below, based on which we plot the dashed line in Figure 1 (**Variance**$_{\text{clean}}$) and dashed gray in Figure 3 (**Bias**$^2$).

$$
\mathbf{Bias}^2 = \frac{1}{4} \Phi_3(\lambda_0, \gamma)^2
$$

$$
\mathbf{Variance}_{\text{clean}} = \begin{cases} \frac{\Phi_1(\lambda_0, \gamma)}{2\Phi_2(\lambda_0, \gamma)} - \frac{(1-\gamma)(1-2\gamma)}{2\gamma} - \frac{1}{4}\Phi_3(\lambda_0, \gamma)^2, & \gamma \leq 1, \\ \frac{\Phi_1(\lambda_0, 1/\gamma)}{2\Phi_2(\lambda_0, 1/\gamma)} - \frac{\gamma-1}{2} - \frac{1}{4}\Phi_3(\lambda_0, \gamma)^2, & \gamma > 1, \end{cases}
$$

where

$$
\begin{aligned}
\Phi_1(\lambda_0, \gamma) =& \lambda_0(\gamma + 1) + (\gamma - 1)^2, \\
\Phi_2(\lambda_0, \gamma) =& \sqrt{(\lambda_0 + 1)^2 + 2(\lambda_0 - 1)\gamma + \gamma^2}, \\
\Phi_3(\lambda_0, \gamma) =& \Phi_2(\lambda_0, \gamma) - \lambda_0 - \gamma + 1.
\end{aligned}
$$

A.3   PROOF OF THEOREM 1

**Lemma 3.** *Define* $\tilde{B} = W^\top (WW^\top + \lambda_0 I)^{-1} W$. $\|\tilde{B}\tilde{B}^\top - \frac{n}{d} AA^\top\|_2 = 0$ *almost surely, i.e.,* $\mathbb{P}(\|\tilde{B}\tilde{B}^\top - \frac{n}{d} AA^\top\|_2 \leq \epsilon) \geq 1 - \delta$ *where $\epsilon$ and $\delta$ both tend to 0 under the asymptotics $n \to \infty$, $d \to \infty$ and $n/d \to \infty$.*

*Proof.* Define

$$\Delta := \frac{d}{n} X X^\top - I,$$
$$\Psi := (WW^\top + \lambda_0 I)^{-1}$$
$$\Gamma := (\frac{d}{n} W X X^\top W^\top + \lambda_0 I)^{-1}$$
$$\Omega := \Gamma - \Psi.$$

Then we have

$$\tilde{B}\tilde{B}^\top - \frac{n}{d} AA^\top = \Omega W \Delta W^\top \Omega + \Psi W \Delta W^\top \Psi + \Omega W \Delta W^\top \Psi + \Psi W \Delta W^\top \Omega$$
$$+ \Omega W W^\top \Omega + \Omega W W^\top \Psi + \Psi W W^\top \Omega.$$

By triangle inequality and the sub-multiplicative property of spectral norm we have

$$\|\tilde{B}\tilde{B}^\top - \frac{n}{d} AA^\top\|_2 \leq \|W\|_2^2 \|\Omega\|_2^2 \|\Delta\|_2 + \|W\|_2^2 \|\Psi\|_2^2 \|\Delta\|_2 + 2\|W\|_2^2 \|\Psi\|_2 \|\Omega\|_2 \|\Delta\|_2$$
$$+ \|W\|_2^2 \|\Omega\|_2^2 + 2\|W\|_2^2 \|\Psi\|_2 \|\Omega\|_2$$

It remains to show that with the asymptotic assumption $n/d \to \infty$, $\|\Omega\|_2 = 0$ and $\|\Delta\|_2 = 0$ almost surely, and $\|W\|_2$ and $\|\Psi\|$ can be bounded from above Yang et al. (2020):

$$\mathbb{P}(\|\Delta\|_2 \leq 4\sqrt{\frac{d}{n}} + 4\frac{d}{n}) \geq 1 - e^{-d/2}$$
$$\|\Omega\|_2 \leq \|\Psi\|_2^2 \|W\|_2^2 \|\Delta\|_2 + O(\|\Delta\|_2)$$
$$\|W\|_2 \overset{a.s.}{=} 1 + \sqrt{\eta} < \infty$$
$$\|\Psi\|_2 \leq \frac{1}{\lambda_0} < \infty.$$

Therefore we have $\|\tilde{B}\tilde{B}^\top - \frac{n}{d} AA^\top\|_2 = 0$ almost surely.   □

**Corollary 4.** $\frac{\sigma^2}{d} \|A\|_F^2 = \frac{\kappa}{d} \|\tilde{B}\|_F^2$ *almost surely.*

*Proof.* By lemma 3 we have

$$|\text{Tr}(\frac{1}{n} \tilde{B}\tilde{B}^\top - \frac{1}{d} AA^\top)| = \frac{1}{n} \text{Tr}(\tilde{B}\tilde{B}^\top - \frac{n}{d} AA^\top)$$
$$\leq \frac{d}{n} \|\tilde{B}\tilde{B}^\top - \frac{n}{d} AA^\top\|_2$$
$$= 0,$$

which yields $\frac{1}{d} \|A\|_F^2 = \frac{1}{n} \|\tilde{B}\|_F^2$ and thus $\frac{\sigma^2}{d} \|A\|_F^2 = \frac{\kappa}{d} \|\tilde{B}\|_F^2$.   □

Now it only remains to compute $\frac{1}{d} \|\tilde{B}\|_F^2$. By Sherman–Morrison formula,

$$\tilde{B} = I - (I + \frac{\alpha}{\eta} Q)^{-1},$$

where $\alpha = \lambda_0^{-1}$, $Q = (d/p) W^\top W$ and $\eta = d/p = 1/\gamma$. Let $F^Q$ be the empirical spectral distribution of $Q$, i.e.,

$$F^Q(x) = \frac{1}{d} \#\{j \leq d : \lambda_j \leq x\},$$

where $\#S$ denotes the cardinality of the set $S$ and $\lambda_j$ denotes the $j$-th eigenvalue of $\boldsymbol{Q}$. Then

$$\frac{\|\tilde{\boldsymbol{B}}\|^2}{d} = \int_{\mathbb{R}^+} \frac{(\frac{\alpha}{\eta}x)^2}{(1+\frac{\alpha}{\eta}x)^2} dF^{\boldsymbol{Q}}(x).$$

By Marchenko-Pastur Law Bai & Silverstein (2010),

$$\frac{\|\tilde{\boldsymbol{B}}\|^2}{d} = \frac{1}{2\pi} \int_{\eta_-}^{\eta_+} \frac{\sqrt{(\eta_+ - x)(x - \eta_-)}(\frac{\alpha}{\eta}x)^2}{\eta x (1+\frac{\alpha}{\eta}x)^2} dx$$

$$= \frac{1}{2\alpha} \left( \frac{-\alpha^2/\eta^2 - (3\alpha - 2\alpha^2)/\eta - \alpha^2 - 3\alpha - 2}{\sqrt{\alpha^2/\eta^2 + (2\alpha - 2\alpha^2)/\eta + \alpha^2 + 2\alpha + 1}} + \alpha/\eta + \alpha + 2 \right). \tag{2}$$

Combining equation 2 and corollary 4 and substituting $\alpha = \lambda_0^{-1}, \eta = \gamma^{-1}$ into the result completes the proof.

### A.4   PROOF OF THEOREM 2

Define the following shorthand

$$\boldsymbol{F} := \boldsymbol{W}\boldsymbol{X}$$
$$\boldsymbol{D}_i := \mathbf{diag}(\boldsymbol{M}_{i,1}, \boldsymbol{M}_{i,2}, \ldots, \boldsymbol{M}_{i,p})$$
$$\mu_i := i\text{-th entry of } \boldsymbol{\mu}$$
$$\boldsymbol{V}_i^\top := i\text{-th row of } \boldsymbol{V}$$
$$\boldsymbol{\Sigma} := \sum_{i=1}^q \mu_i^2 \boldsymbol{D}_i$$
$$\boldsymbol{H} := \boldsymbol{y}^\top \boldsymbol{F}^\top \boldsymbol{\Sigma} \boldsymbol{F} (\boldsymbol{F}^\top \boldsymbol{\Sigma} \boldsymbol{F} + \lambda \mathbf{I})^{-1}$$

The parameter of the second layer is given by ridge regression:

$$\hat{\boldsymbol{V}} = \underset{\boldsymbol{V} \in \mathbb{R}^{q \times p}}{\arg\min} \| ((\boldsymbol{V} \odot \boldsymbol{M})\boldsymbol{W}\boldsymbol{X})^\top \boldsymbol{\mu} - \boldsymbol{y}\|_F^2 + \lambda \|\boldsymbol{V}\|_F^2$$

$$= \underset{\boldsymbol{V} \in \mathbb{R}^{q \times p}}{\arg\min} \| \sum_{i=1}^q \mu_i \boldsymbol{V}_i^\top \boldsymbol{D}_i \boldsymbol{F} - \boldsymbol{y}^\top \|_F^2 + \lambda \|\boldsymbol{V}\|_F^2.$$

Solving the above yields

$$\hat{\boldsymbol{V}}_i^\top = \frac{1}{\lambda}(\boldsymbol{y}^\top - \boldsymbol{H})\boldsymbol{F}^\top(\mu_i \boldsymbol{D}_i).$$

Given a clean test example $(\boldsymbol{x}, y)$, the expression of the risk is

$$\mathbf{Risk} = \mathbb{E}\| \sum_{i=1}^q \mu_i \hat{\boldsymbol{V}}_i^\top \boldsymbol{W}\boldsymbol{x} - \boldsymbol{\theta}^\top \boldsymbol{x}\|_F^2$$

$$= \mathbb{E}\| \frac{1}{\lambda}(\boldsymbol{y}^\top - \boldsymbol{H})\boldsymbol{F}^\top \boldsymbol{\Sigma} \boldsymbol{W}\boldsymbol{x} - \boldsymbol{\theta}^\top \boldsymbol{x}\|_F^2$$

$$= \mathbb{E}\| \frac{1}{\lambda}(\boldsymbol{y}^\top - \boldsymbol{y}^\top \boldsymbol{F}^\top \boldsymbol{\Sigma} \boldsymbol{F}(\boldsymbol{F}^\top \boldsymbol{\Sigma} \boldsymbol{F} + \lambda \mathbf{I})^{-1})\boldsymbol{F}^\top \boldsymbol{\Sigma} \boldsymbol{W}\boldsymbol{x} - \boldsymbol{\theta}^\top \boldsymbol{x}\|_F^2$$

Observe that the diagonal matrix $\boldsymbol{\Sigma}$, whose diagonal entries $\sum_{i=1}^q \mu_i^2 \boldsymbol{M}_{i,j} = \frac{1}{q}\sum_{i=1}^q (\sqrt{q}\mu_i)^2 \boldsymbol{M}_{i,j}$ converge in probability to $\alpha$ as $q \to \infty$, captures all the effects of $\boldsymbol{\mu}$ and $\boldsymbol{M}$ on the risk. Thus we can replace $\boldsymbol{\Sigma}$ with $\alpha \mathbf{I}_p$

$$\mathbf{Risk} = \mathbb{E}\| \frac{\alpha}{\lambda}\boldsymbol{y}^\top \boldsymbol{F}\boldsymbol{W}\boldsymbol{x} - \frac{\alpha^2}{\lambda^2}\boldsymbol{y}^\top \boldsymbol{F}^\top \boldsymbol{F}(\frac{\alpha}{\lambda}\boldsymbol{F}^\top \boldsymbol{F} + \mathbf{I})^{-1}\boldsymbol{F}^\top \boldsymbol{W}\boldsymbol{x}\|_F^2. \tag{3}$$

It remains to show that the expression of the risk is exactly the same as the risk in the setting of Section 3.1 with $\lambda$ replaced by $\lambda/\alpha$. Write down the risk in Section 3.1 (for convenience denote it by $\mathbf{Risk}_0$):

$$\mathbf{Risk}_0 = \mathbb{E}\|\boldsymbol{y}^\top \boldsymbol{F}^\top(\boldsymbol{F}\boldsymbol{F}^\top + \lambda \mathbf{I})^{-1}\boldsymbol{W}\boldsymbol{x} - \boldsymbol{\theta}^\top \boldsymbol{x}\|_F^2$$

$$= \mathbb{E}\| \frac{1}{\lambda}\boldsymbol{y}^\top \boldsymbol{F}\boldsymbol{W}\boldsymbol{x} - \frac{1}{\lambda^2}\boldsymbol{y}^\top \boldsymbol{F}^\top \boldsymbol{F}(\frac{1}{\lambda}\boldsymbol{F}^\top \boldsymbol{F} + \mathbf{I})^{-1}\boldsymbol{F}^\top \boldsymbol{W}\boldsymbol{x}\|_F^2. \tag{4}$$

Equation 4 is obtained by applying Woodbury matrix identity to $(\boldsymbol{F}\boldsymbol{F}^\top + \lambda \mathbf{I})^{-1}$. It is easy to check that replacing $\lambda$ in RHS of equation 4 with $\lambda/\alpha$ yields the same as RHS of equation 3.

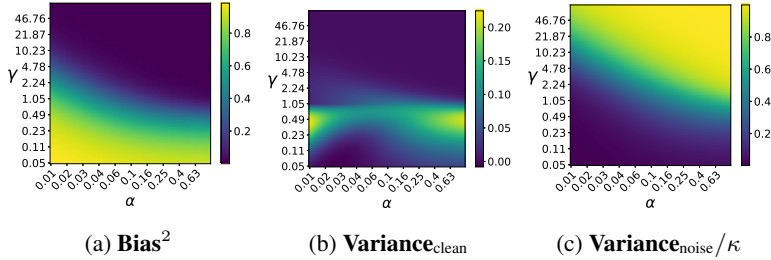

(a) **Bias**$^2$    (b) **Variance**$_{\text{clean}}$    (c) **Variance**$_{\text{noise}}/\kappa$

Figure 11: Expressions of **Bias**$^2$, **Variance**$_{\text{clean}}$ and **Variance**$_{\text{noise}}/\kappa$ in Theorem 2

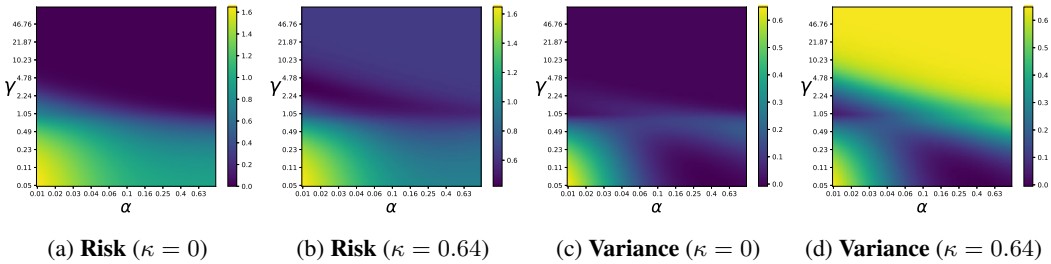

(a) **Risk** ($\kappa = 0$)    (b) **Risk** ($\kappa = 0.64$)    (c) **Variance** ($\kappa = 0$)    (d) **Variance** ($\kappa = 0.64$)

Figure 12: Expressions of the risk and variance under different noise levels with $\tau = \gamma$. We let $\lambda_0 = 0.05$.

## A.5   **BIAS**$^2$ AND **VARIANCE**$_{\text{NOISE}}/\kappa$ IN SECTION 3.2

In Figure 11 we plot the expressions of **Bias**$^2$ and **Variance**$_{\text{noise}}/\kappa$ based on Theorem 2. **Bias**$^2$ monotonically decreases along both axes and **Variance**$_{\text{noise}}/\kappa$ monotonically increases along both axes. **Variance**$_{\text{clean}}$ variance is unimodal along y-axis (width), manifests more complicated behavior along x-axis (density), and decreases along both axes once width is sufficiently large, and thus it differs from the variance in classical bias-variance tradeoff.

## A.6   VARIANTS OF THEOREM 2

In Theorem 2 we let $\boldsymbol{\mu}$'s entries be drawn from $\mathcal{N}(0, 1/q)$ so that $q$ does not appear in the expression of the risk. Alternatively we can let $\boldsymbol{\mu}$'s entries be drawn from $\mathcal{N}(0, 1/d)$ and assume $q/d = \tau$. Then the risk and its decomposition are dependent on $\gamma, \tau, \alpha$. Similar to the proof in A.4, we can show that in this case one just need to replace $\lambda_0$ in the setting of Theorem 1 with $\lambda_0/(\tau\alpha)$ to get the expressions of risk. We further let $\tau = \gamma$ and plot the result in Figures 12 and 13.

## A.7   CONNECTION TO THE OBSERVATION MADE BY GOLUBEVA ET AL. (2020)

Our Theorem 2 confirms the observation in Golubeva et al. (2020) that increasing width while keeping the number of parameters fixed (via reducing density) benefits generalization. This disentangles the benefit of increasing width from the effect of enlarging model capacity. In our setting, the expected number of parameters is in proportion to $\gamma\alpha$. We fix $\gamma\alpha$ and plot the risk curve with varied $\gamma$ ($s.t.\alpha \leq 1$) in Figure 14. When $\kappa = 0$ the test loss decreases with width. However, large noise here can still change the shape of the curve. When $\kappa = 0.64$, the test loss is either U-shaped or increasing.

## B   DETAILS OF EXPERIMENTAL SETTING

All experiments are implemented using PyTorch. We use eight Nvidia A40 to run the experiments.

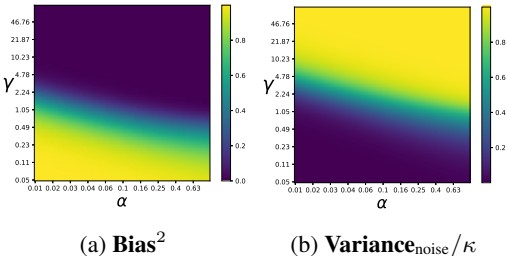

(a) **Bias**$^2$        (b) **Variance**$_{\text{noise}}/\kappa$

Figure 13: Expressions of **Bias**$^2$ and **Variance**$_{\text{noise}}/\kappa$ with $\tau = \gamma$.

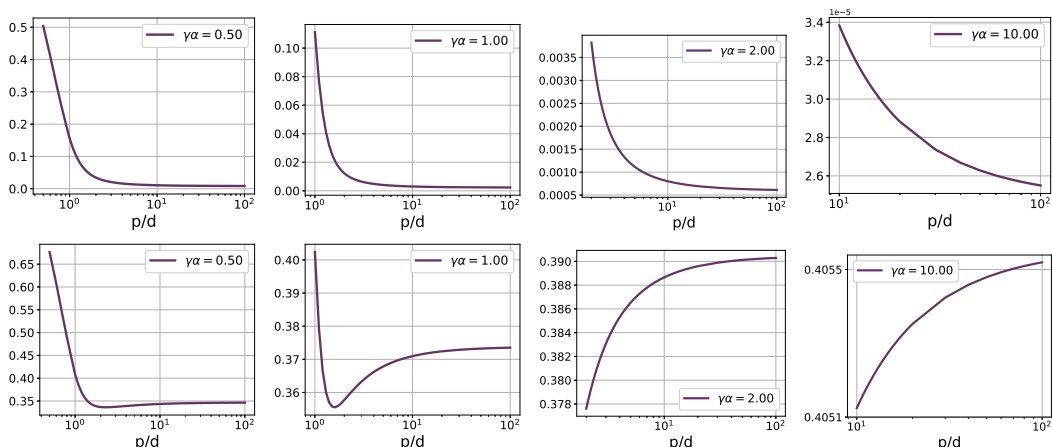

Figure 14: Risk curve with fixed $\gamma\alpha$. **Top:** $\kappa = 0$, the test loss decreases with width. **Bottom:** $\kappa = 0.64$, the test is either U-shaped or increasing.

## B.1 NOISE TYPES

**Symmetric noise**    Symmetric noise is generated by randomly assigning labels to a certain fraction of examples.

**Asymmetric noise**    Asymmetric noise is class-dependent. We follow the scheme proposed in Patrini et al. (2017) which is also widely used in robust method papers (e.g., ELR Liu et al. (2020) and DivideMix Li et al. (2020)). For CIFAR-10, labels are randomly flipped according to the following map: TRUCK→AUTOMOBILE, BIRD→AIRPLANE, DEER→HORSE, CAT→DOG, DOG→CAT. For CIFAR-100, since the 100 classes are grouped into 20 superclasses, e.g. AQUATIC MAMMALS contains BEAVER, DOLPHIN, OTTER, SEAL and WHALE, we flip labels of each class into the next one circularly within super-classes. For both datasets, the fraction of mislabeled examples in the training set is the noise level.

**Web Noise**    Red Stanford Car Jiang et al. (2020) contains images crawled from web. There are 10 different noise levels {0%, 5% 10%, 15%, 20%, 30%, 40%, 60%, 80%} for this dataset and we choose 40% and 80%. For each noise level, the mislabeled web images can only be downloaded from the provided URLs. The training splits and labels are in provided files [1]. The original dataset size is 8144. For 80% noise, there are 6469 web images. However, 590 of the URL links are not functional and among the downloaded JPG files 1871 are corrupted/unopenable, hence we end up with 1675 clean examples and 4008 noisy examples, i.e., the actual noise level is 70.53%. For 40% noise, there are 3241 web images with 313 non-downloadable and 963 not unopenable. Therefore the actual noise level is 29.03%.

---

[1]See their webpage for details `https://google.github.io/controlled-noisy-web-labels/download.html`

## B.2 BASIC EXPERIMENTS

**MNIST and CIFAR-10 with symmetric noise** On MNIST, we train a two-layer ReLU network for 200 epochs using SGD with batch size 64, momentum 0.9, initial learning rate 0.1, a weight decay of $5 \times 10^{-4}$, a learning rate decay of 0.1 every 50 epochs. On CIFAR-10 We train ResNet34 for 1000 epochs using SGD with batch size 128, momentum 0.9, initial learning rate 0.1, weight decay of $5 \times 10^{-4}$, learning rate decay of 0.1 every 400 epochs. For width $w$, there are $w, 2w, 4w, 8w$ filters in each layer of the four Residual Blocks, respectively. We use MSE loss in both settings.

**CIFAR-10/100 with asymmetric noise** On CIFAR-10, we train ResNet34 for 500 epochs using SGD with batch size 128, momentum 0.9, initial learning rate 0.1, weight decay of $5 \times 10^{-4}$, learning rate decay of 0.1 every 100 epochs. On CIFAR-100, we train ResNet34 for 1000 epochs using SGD with batch size 128, momentum 0.9, initial learning rate 0.1, weight decay of $5 \times 10^{-4}$, learning rate decay of 0.1 every 200 epochs. We use Cross Entropy loss in both settings.

**Red Stanford Car** On Red Stanford Car Jiang et al. (2020) with 80% noise. We train InceptionResNet-v2 Szegedy et al. (2017) for 160 epochs with an initial learning rate of 0.1, and a weight decay of $1 \times 10^{-5}$ using SGD and momentum of 0.9 with a batch size of 32. We anneal the learning rate by a factor of 10 at epochs 80 and 120, respectively. We use Cross Entropy loss.

## B.3 VARYING SAMPLE SIZE

**MNIST** The setup is the same as in B.2.

**CIFAR-10/100** On both datasets the noise level is 60%. On CIFAR-10, we train ResNet34 for 1000 epochs using SGD with momentum 0.9, initial learning rate 0.1, weight decay of $5 \times 10^{-4}$, learning rate decay of 0.1 every 100 epochs. The batch size is 128 for sample size 50000, 64 for sample size 25000, 10000, 5000, and 32 for sample size 2500. We set batch size this way to make sure the model can converge and the largest few models can fit each of the training set (e.g., when trained with batch size 128 on 2500 examples even the largest model does not fit the training set; when trained with batch size 64 on 50000 examples the model converges too slowly). On CIFAR-100, we train ResNet34 for 500 epochs using SGD with momentum 0.9, initial learning rate 0.1, weight decay of $5 \times 10^{-4}$, learning rate decay of 0.1 every 100 epochs.

## B.4 EXPERIMENTS WITH ROBUST METHODS

ELR leverages the early learning phenomenon where the network fits clean examples first and then mislabeled examples. It hinders learning wrong labels by regularizing the loss with a term that encourages the alignment between the model's prediction and the running average of the predictions in previous rounds. DivideMix dynamically discards labels that are highly likely to be noisy and trains the model in a semi-supervised manner. For both ELR and DivideMix we use the same setup in the original papers. Liu et al. (2020); Li et al. (2020).

**ELR** We train ResNet-34 using SGD with momentum 0.9, a weight decay of 0.001, and a batch size of 128. The network is trained for 120 epochs on CIFAR-10 and 150 epochs on CIFAR-100. The learning rate is 0.02 at initialization, and decayed by a factor of 100 at epochs 40 and 80 for CIFAR-10 and at epochs 80 and 120 for CIFAR-100. We use 0.7 for the temporal ensembling parameter, and 3 for the ELR regularization coefficient.

**DivideMix** We train ResNet-34 for 200 epochs using SGD with batch size 64, momentum 0.9, initial learning rate 0.02, a weight decay of $5 \times 10^{-4}$, and a learning rate decay of 0.1 at epoch 150. We use 150 for the unsupervised loss weight.

# C ADDITIONAL PLOTS FOR SECTIONS 4 TO 6

## C.1 BASIC EXPERIMENTS: VARYING WIDTH

**Bias**$^2$ and test accuracy on MNIST and CIFAR-10 are shown in Figure 15. Test accuracy on CIFAR-10/100 with asymmetric noise is shown in Figures 16a and 16b. We also plot test loss and test accuracy of models trained on CIFAR-10 with 30% asymmetric noise in Figures 16c and 16d. Here

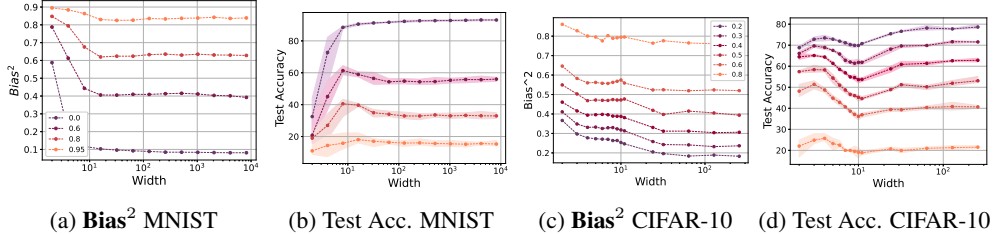

(a) **Bias**$^2$ MNIST  (b) Test Acc. MNIST  (c) **Bias**$^2$ CIFAR-10  (d) Test Acc. CIFAR-10

Figure 15: **Bias**$^2$ and test accuracy on MNIST and CIFAR-10.

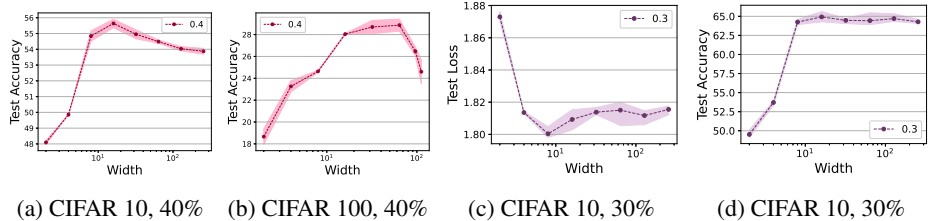

(a) CIFAR 10, 40%  (b) CIFAR 100, 40%  (c) CIFAR 10, 30%  (d) CIFAR 10, 30%

Figure 16: We plot test accuracy on CIFAR-10 (a) and CIFAR-100 (b) under 40% asymmetric noise. We also plot test loss (c) and test accuracy (d) on CIFAR-10 under 30% asymmetric noise.

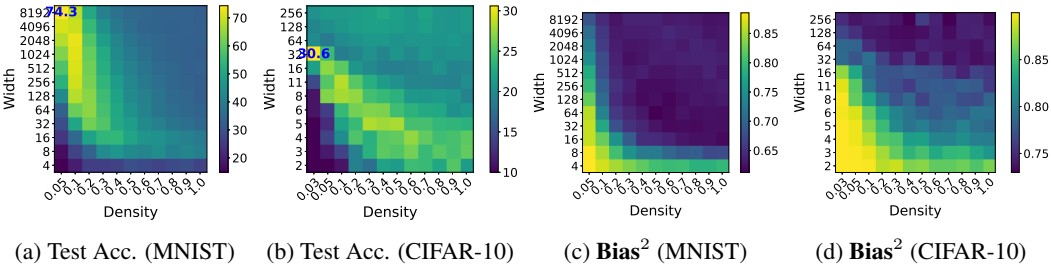

(a) Test Acc. (MNIST)  (b) Test Acc. (CIFAR-10)  (c) **Bias**$^2$ (MNIST)  (d) **Bias**$^2$ (CIFAR-10)

Figure 17: Test accuracy and bias plotted with varied width and density.

we can still observe a slight ascent from width 128 to 256. As expected, the final ascent in this setting is less pronounced than under 40% noise (Figure 5c).

## C.2  BASIC EXPERIMENTS: JOINT EFFECT OF WIDTH AND DENSITY

We plot test accuracy and bias in Figure 17. The bias decreases when density and width increase. In both settings, the optimal accuracy is achieved by models with smallest density.

## C.3  SMALLER DENSITY VS. STRONGER $l_2$ REGULARIZATION

In Theorem 2, we show that reducing density is equivalent to increasing the strength of $l_2$ regularization. To see whether this applies to more complex settings, we train ResNet34 at width 16 on CIFAR-10 with 50% symmetric noise. We vary weight decay and network density. Results are shown in Figure 18. Both loss curves are U-shaped, but reducing density can achieve a lower loss/higher accuracy than increasing $l_2$ regularization. The non-equivalence itself is not unexpected given the non-linearity in neural networks and how dropping a large amount of connection can significantly change the training dynamics of SGD. However, it would be interesting to investigate why reducing density is better and whether it is the same case in other settings.

## C.4  VARYING SAMPLE SIZE

We plot test accuracy with varied sample size in Figure 19.

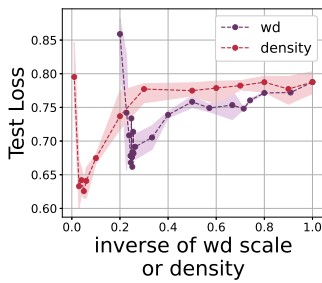 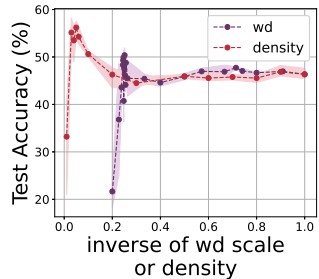

Figure 18: When varying weight decay, the density is fixed to 1. When varying density, the weight decay is fixed to $0.0005$. The x-axis represents the inverse of the scale-up of weight decay, i.e., the current weight decay divided by $0.0005$, for the weight decay curve (purple), and model density for the density curve (red).

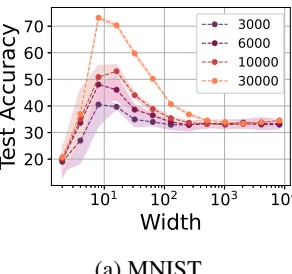

(a) MNIST

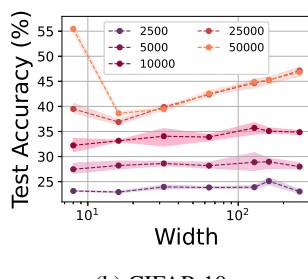

(b) CIFAR 10

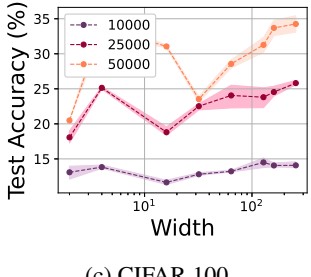

(c) CIFAR 100

Figure 19: Test accuracy on MNIST and CIFAR-10/100 with varied sample size.

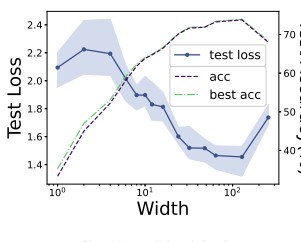

(a) CIFAR 10, 80% sym

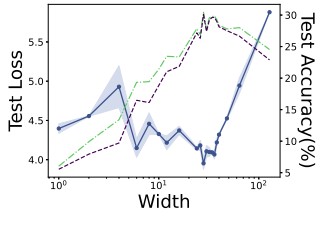

(b) CIFAR 100, 80% sym

Figure 20: Test performance of models at increasing width trained with ELR on CIFAR-10/100.

### C.5 MORE RESULTS FOR MODELS TRAINED WITH ROBUST METHODS

In addition to the experiments presented in the main paper, we vary width of models trained with ELR under 80% noise and vary density

We show the results of the following experiments on CIFAR-10/100: width experiments with ELR under 80% noise 20, density experiments with ELR 21, density experiments with DivideMix under 60% noise 22. We also show the result of density experiments on Red Stanford Car (70% noise) where we train InceptionResNet-v2 with ELR 23. In Figures 20 and 21, 'best_acc' refers to the highest test accuracy observed during training, which is usually reported in robust method papers.

## D  MEASURING COMPLEXITY OF LEARNED FUNCTIONS

We consider three measures:

- Trajectory length of the first layers bias. $\sum_{t \in T} \frac{\|\boldsymbol{b}_1^{(t+1)} - \boldsymbol{b}_1^t\|_2}{\alpha_t \epsilon_{f^{(t)}}}$ where $T$ is a set of iteration indices, $\boldsymbol{b}_1^{(t)}$ is the parameter of the first layer bias at iteration $t$, and $\epsilon_{f^{(t)}}$ is the gradient

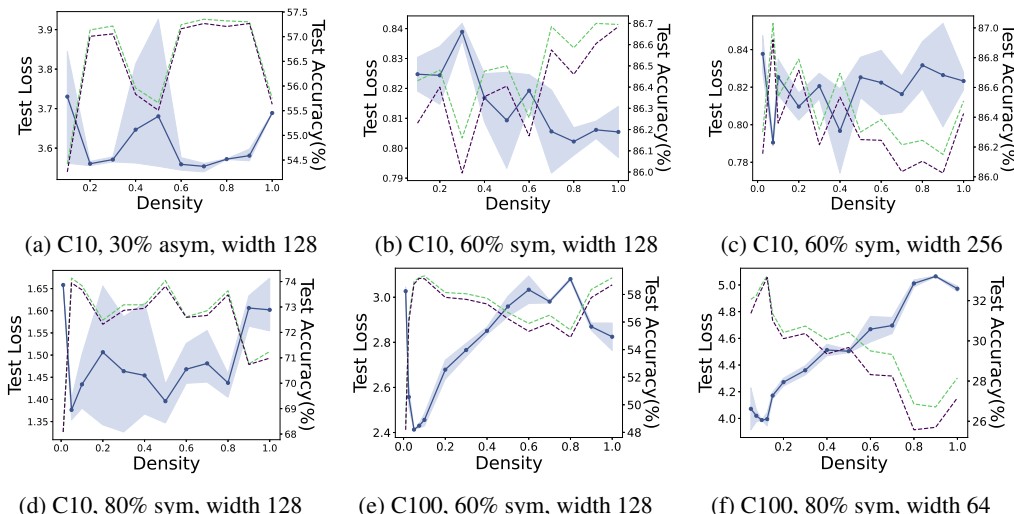

(a) C10, 30% asym, width 128    (b) C10, 60% sym, width 128    (c) C10, 60% sym, width 256

(d) C10, 80% sym, width 128    (e) C100, 60% sym, width 128    (f) C100, 80% sym, width 64

Figure 21: Test performance of models at increasing density trained with ELR on CIFAR-10/100.

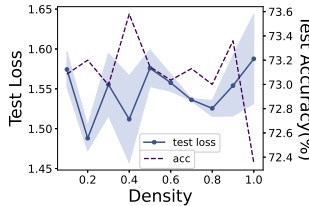

Figure 22: Test performance of models at increasing density trained with DivideMix on CIFAR-100. We set the width to 64.

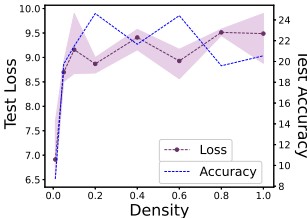

Figure 23: Test performance of models at increasing density trained with ELR on Red Stanford Car.

of the loss w.r.t. the network's output at epoch $t$. Loukas et al. (2021) shows that, under certain conditions, the above can be both upper and lower bounded in terms of the Lipschitz constants of functions represented by the network at iterations in $T$ (see their Theorem 1). This implies that the first layer bias travels longer during training when it is fitting a more complex function.

- Variance of the first layer bias $\mathrm{avg}_{t \in T} \| \boldsymbol{b}_1^{(t)} - \mathrm{avg}_{t \in T} \boldsymbol{b}_1^{(t)} \|_2^2$. Loukas et al. (2021) also provides a lower bound for the above in terms of the Lipschitz constant and $\epsilon_{f^{(t)}}$ (their Corollary 2). Thus a larger Lipschitz constant leads to a higher lower bound, meaning that when fitting a lower complexity function, the network's bias will update more frequently during training.

- Product of spectral norms of the layer parameters. This is known as an upper bound for the network's Lipschitz constant Szegedy et al. (2013). For convolutional layers, the spectral norm is computed using the FFT-based algorithm in Sedghi et al. (2018).

Our setting the same as in Loukas et al. (2021) (Task 2 in Section 6). We train a CNN on CIFAR-10 DOG vs AIRPLANE. The CNN has one identify layer, 2 convolutional layers with kernel size 5, a

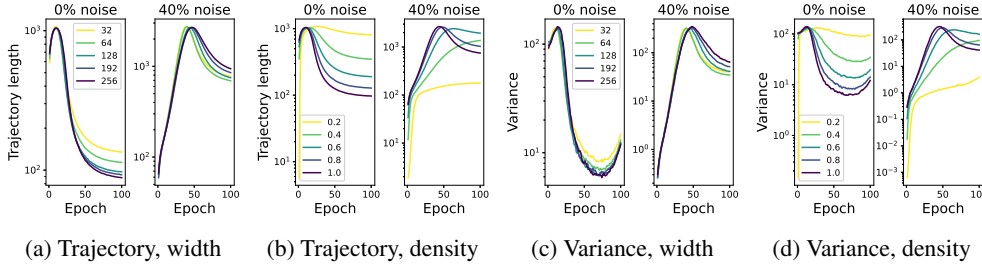

Figure 24: Trajectory length and variance of the first layer bias during each epoch. We vary both width and density. When varying the density we fix the width to 128.

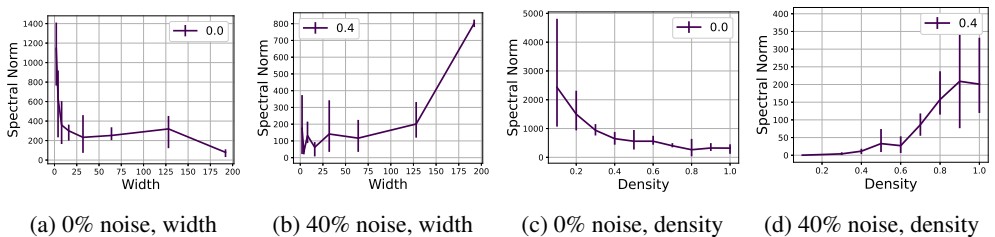

Figure 25: When width or density increases, spectral norm decreases under 0% noise but increases under 40% noise.

fully connected ReLu layer with size 384 and a linear layer. The width is controlled by the number of convolutional channels. For width $w$ there are $16w$ channels in each convolutional layer. We use BCE loss and train the models for 200 epochs using vanilla SGD with batch size 1. We use exponential learning rate decay with a factor $10^{-50}$. The trajectory length and variance are computed every epoch, i.e., we let $T$ be the iterations of each SGD epoch. Thus they capture the complexity of the function represented by the CNN at every epoch. The results are shown in Figure 24. We see that noise does change the relative complexity of functions learned by models at increasing width/density. The pattern is more clear for width (Figures 24a and 24c) where both quantities decrease under 0% noise, and increase under 40% noise, when width increases. In Figure 25 we plot the product of layer-wise spectral norms of the neural network at the last epoch and it shows a similar pattern. These results suggest that the originally negative correlation between size and complexity/smoothness can be flipped by label noise.

## E    POTENTIAL CONNECTION TO BENIGN OVERFITTING

Benign overfitting refers to the phenomenon where models trained to overfit the training set still achieve nearly optimal generalization performance Bartlett et al. (2020); Tsigler & Bartlett (2020); Chatterji et al. (2021); Cao et al. (2022); Frei et al. (2022); Mallinar et al. (2022). Recently Cao et al. (2022) shows that sufficiently large models overfit benignly if the product between sample size and signal-to-noise ratio (SNR) is large, and catastrophically otherwise. Interestingly, this is the same condition for the final ascent to not occur in our theoretical setting (Section 3, the SNR in our setting is $1/\sigma^2$). Therefore one can explain our result from the perspective of benign overfitting: when the condition is satisfied, a *sufficiently large* model overfits benignly, and therefore increasing model size toward sufficiently large can possibly help generalization. That said, our result has broader implications because it (1) compares models at arbitrary size that may not be sufficiently large and (2) does not restrict to the benign overfitting regime. Indeed, neural networks along with many other real interpolating methods are in the *tempered overfitting* regime Mallinar et al. (2022). Our result suggests that the condition for benign overfitting identified in Cao et al. (2022) can be possibly extended to judge the relative *benignity* of tempered overfitting at different model sizes. We see theoretically establishing the above as an important direction of future work.

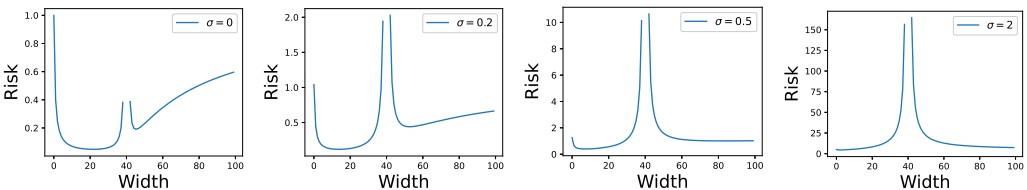

Figure 26: We plot the risk as a function of the width for the 'prescient' selection model using Theorem 2.1 in Belkin et al. (2020). We see that as noise increases, the risk in the overparameterized regime turns from a U-shaped into a descent.

## F NON-ASYMPTOTIC RESULTS FROM PREVIOUS WORK

We clarify here that Theorem 1 cannot be obtained from taking the limit of previous work. The setting in our paper differs from previous ones that aim at deriving non-asymptotic results. We chose this setting as it allows incorporating the three important sources of randomness, namely training data, label noise, and random features that mimics the randomness in initialization and optimization (as opposed to e.g. models with fixed design or without random feature), and allows deriving interpretable closed form results (as opposed to non-linear feature or n/d=const limit). We are not aware of any non-asymptotic result that gives explicit expression of the bias-variance tradeoff, as we derived in this setting. We give a few examples that the final ascent has not been reflected in previous theoretical models on double descent: Belkin et al. (2020): Corollary 2.2 (Gaussian model with random selected features) doesn't capture the final ascent since the noise dependent term always decreases with the width. The 'prescient' selection model (their Fig 2) has an ascent in the end, however, as we plotted in Figure 26, (1) the ascent exists even without noise, thus the result doesn't show the double descent phenomenon in the first place, and (2) the ascent diminishes as we increase noise, and therefore shows an effect opposite to our empirical observation. The Fourier series model in their Theorem 3.1 used to show double descent is for noise free setting. Hastie et al. (2019): Among the models they considered, only the minimum-norm least squares with isotropic features can manifest a U-shaped curve in the overparameterization regime (their Fig 1). However, the effect of noise here is exactly the opposite of what we observed. As is described in their Sec 3.2, in the overparameterization regime, the loss monotonically decreases when SNR<1 (noise is large) and has a local minimum when SNR>=1 (noise is small).

