# OpenReview forum: "The Final Ascent: When Bigger Models Generalize Worse on Noisy-Labeled Data"
_ICLR.cc/2023/Conference — Submitted to ICLR 2023_

### Official Review · Reviewer_RtYb · 2022-10-24

**Confidence:** 4
**Correctness:** 4
**Technical Novelty And Significance:** 2
**Empirical Novelty And Significance:** 3
**Recommendation:** 5

**Clarity, Quality, Novelty And Reproducibility:**

The paper is overall well-written and the results are clear.  I found the
theoretical results to be a bit lacking in novelty and originality.


**Strength And Weaknesses:**

Strengths

1. Studying the effect of overparameterization in noisy settings is
a well-motivated and natural question. This paper provides some theoretical and
empirical results and also some observations about overparameterization in
conjunction with robust training techniques in noisy settings that fit in this
line of research.

Weaknesses


1. The bias-variance decomposition closely follows the proofs of the prior work
of Yang et al. [1].  The theoretical results on the 2-layer linear network
(Theorem 1) seems to follow by using the same techniques as in [1]. Overall, I
found the theoretical results presented in this work not particularly
technically challenging or novel as they follow closely the model and proofs of
[1].

2. There is a discrepancy between the noise models in the theoretical and
experimental results.  The theoretical results consider a regression setting
with additive gaussian noise and the experimental results consider
classification settings with symmetric or asymmetric random classification
noise.  I agree with the authors that studying label noise in classification
settings is more relevant so I think it would be better if the theoretical
results also considered a similar setting.

3. In the experimental results the noise rates required so that the ascent in
test loss happens seem to be very large (in most cases above 50%).  For
example, in MNIST the noise rates are 80% and 95%.  For such large noise rates
in the training data, it does not seem very surprising that the test loss is
eventually increasing as the noisy examples are the majority.  I think it would
be interesting to also consider adding random noise to the test data and see
whether the "last ascent" would then turn into a "descent".

[1] Zitong Yang, Yaodong Yu, Chong You, Jacob Steinhardt, and Yi Ma. Rethinking bias-variance trade-off for generalization of neural networks. In International Conference on Machine Learning, pp. 10767–10777. PMLR, 2020.


Typos and Minor Issues

Abstract, "particularly when trained on smaller dataset" -> datasets

Theorem 1, "$n/d = \infty$ should be $n/d \to \infty$ (The same for Theorem 2).



**Summary Of The Paper:**

This paper considers the effect of label noise on generalization when training
overparameterized models.  Prior work has given theoretical and empirical
evidence that overparameterization improves generalization.  This work shows
that, when the labels of the training data are very noisy, overparameterization
may actually harm generalization and provides a theoretical analysis of the
phenomenon for 2-layer linear networks.  They provide empirical evidence for
this effect using both synthetic and real noise for classification tasks.  They
also consider using recent robust training methods (DivideMix and ELR) and find
that in some cases using these methods may actually exacerbate the phenomenon
in the sense that it appears for lower noise rates.  They conclude that using
larger models may not always be the right choice in the presence of label noise
and using robust training methods does not seem to make things better.


**Summary Of The Review:**


This paper studies an interesting and well-motivated setting and presents some
theoretical and empirical results.  I believe that there is some value in the
extensions of the theoretical results of [1] in the noisy settings and
empirical observations on using robust training methods are interesting.
However, the theoretical results presented in this work seem to heavily rely on
prior work and do not seem particularly surprising and novel.  I think it would
be be more interesting if the noise models in the theoretical framework were
more aligned with the experimental setting (random classification noise).
Moreover, I am not sure on the practical significance of the empirical results
as considering that the majority of the training data are corrupted seems to be
a bit unrealistic.  Therefore, overall, I am not sure whether this work is
above the threshold for ICLR.

---

> ### Author Response · Authors · 2022-11-14
> **Response to Reviewer RtYb**
>
> ### Novelty of theoretical results
> We believe that our results are both important and novel. Although the analysis is built upon Yang's, they didn’t consider label noise in the analysis therefore did not observe the final ascent. Instead we systematically study this phenomenon and provide a straightforward theoretical explanation. Moreover, our analysis incorporates the effect of density on the generalization (Theorem 2), *for the first time*. Originally the layer parameter is a vector and reducing width and reducing density are the same. To analyze the effect of density, we added an extra layer to the network so that the trainable layer has multiple output dimensions (considering layer parameters as a matrix, reducing width deletes rows while reducing density deletes individual entries). By analyzing the minimizer of the regression loss, we showed that the mask applied to the trainable parameters scales down the random features in the limit. Hence, reducing density effectively scales up the l2 regularization parameter (in Sec 4.3 we further discuss that reducing density may have an effect beyond regularization in neural networks). The result also shows that reducing density has additional benefits in the sense that the optimal width at a smaller density achieves better generalization than the optimal width at a larger density (Figures 4(b)(d)). To the best of our knowledge, we are the first to present this phenomenon and it could help understand the relation between model capacity and generalization. Meanwhile it provides theoretical support for [2]’s empirical observation in the noiseless setting that, when the total number of trainable parameters is fixed, using a sparser but wider network further improves generalization. All of these make our analysis distinct from the recent theoretical studies on overparameterization that mostly focuses on the effect of width.
>
> ### Regression vs classification
> To characterize the final ascent in modern machine learning, the analysis must successfully capture double descent without noise in the first place. The double descent phenomenon has only been *theoretically characterized* in regression tasks although it has been empirically observed in classification tasks. For this reason we took ridge regression for our theoretical analysis. However, as we showed, our theory can indeed give us useful insights and match the empirical observations in classification tasks (e.g., Fig 3 shows that our theory can explain the shape of the variance curve). We thank the reviewer for bringing this up, and agree that theoretically characterizing double descent and other related phenomena in classification problems can be an interesting direction for future work.
>
> ### Noise ratio required for the final ascent to occur
> We emphasize that while a relatively high **symmetric label noise** is required to observe the final ascent, a **moderate noise** level results in the final ascent under the following *more realistic* scenarios: (1) when noise is **asymmetric** (Fig 5, Fig 9(a) and Fig 16), or when (2) **robust methods** are applied (Fig 9). We expect the phenomenon to be even more pronounced under stronger types of noise, i.e., manual-labeling mistakes (usually class/instance-dependent and more skewed than symmetric label flipping) or adversarial noise.
>
> We note that adding noise to the test set does not change the picture. If noisy labels are randomly generated, it is easy to theoretically show that in regression tasks (e.g, in our model), adding noise to the test set would only lead to an additional constant term (irreducible error) in the test loss. And in general no matter what the model learns from the training set, it can never predict the random labels on the test set. If the label noise has a particular structure, e.g., class dependent, then adding the same noise to the test set (e.g., on both training and test sets images of cats are labeled as dog) can turn the ascent into descent. However, in this case the test set is problematic and the test loss is no longer a measure of generalization.
>
> [2] Golubeva, Anna, Behnam Neyshabur, and Guy Gur-Ari. "Are wider nets better given the same number of parameters?." arXiv preprint arXiv:2010.14495 (2020).

---

> ### Author Response · Authors · 2022-12-06
> **Looking forward to the reviewer response**
>
> Dear reviewer,
>
> We would like to follow up to see if our response addressed your concerns and if there is anything else we can clarify before the deadline.

---

### Official Review · Reviewer_fFmQ · 2022-10-24

**Confidence:** 3
**Correctness:** 3
**Technical Novelty And Significance:** 2
**Empirical Novelty And Significance:** 2
**Recommendation:** 3

**Clarity, Quality, Novelty And Reproducibility:**

- How is the variance decomposition (Eq. 1) related to the decomposition of Adlam and Pennington (2022a) in Example 2?
- Can Theorem 1 be obtained from previous work by taking the limit \psi \to \infty?
- Taking psi \to \infty while keeping kappa finite means \sigma^2 goes to infinity. This is a very odd limit. Doesn't this mean the noise is much larger than the signal?
- Page 6: There are bias-variance decompositions for losses other than MSE see "A Generalized Bias-Variance Decomposition for Bregman
Divergences" Pfau (2013).

**Strength And Weaknesses:**

**Strengths**
- The paper provides closed-form solutions for the bias and variance.
- There are experiments on real neural network architectures.

**Weaknesses**
- There is a lack of novelty. The main theoretical results are a minor extension of Yang et al., and it's unclear whether they could just be obtained by taking a limit of previously published results.
- The analysis is restricted to linear random features. Why is this restriction necessary?

**Summary Of The Paper:**

The paper studies the effect of label noise on the test error of overparameterized models. In particular, it shows that when the labels are sufficiently noisy, the test error increases again with model size (after the double descent peak). The authors show this theoretically in a simple linear random features model, and provide empirical evidence for it in real neural networks.

**Summary Of The Review:**

The main findings of the paper lack novelty, which could hurt the paper's interest to the ICLR community. In addition, the relationship of the theoretical results to previous work is not fully explained or explored.

---

> ### Author Response · Authors · 2022-11-14
> **Response to Reviewer fFmQ, Part 2**
>
> ### Non-linear random feature
> For non-linear random features, it is technically difficult to derive a closed form expression especially when considering random design. Analysis for non-linear features can be found in (Adlam 2022a) Lemma 1 and Theorem 1, where the bias and variance are functions of (derivatives of) solutions to coupled polynomial equations. And the exact effect of the ridge parameter, noise ratio and width remains unclear. In contrast, our theoretical analysis provides an interpretable and straightforward illustration of the final ascent phenomenon, for which we consider the linear case. We expect more advanced theories can be developed to explicitly characterize our phenomenon in non-linear settings.
>
> ### Does the scale in the theory mean the noise is much larger than the signal?
> Yes. However, we note that this is because of our scale n/d -> inf, which we take to simplify the analysis. Note that the strength of regularization also scales with n/d, which means noise should also take the same scale otherwise its effect vanishes. We hope for future work the final ascent can be characterized in the case where n/d=const., and in this case the noise-to-signal ratio isn’t necessarily large. We also emphasize that our theory does match our empirical observation (Fig 1) and provides a straightforward explanation for the final ascent.
>
> ### Variance decomposition (Eq. 1) and the decomposition of (Adlam 2022a) in Example 2?
> We decompose the variance into two terms $V_{clean}$ and $V_{noise}$. Essentially, $V_{clean}$ is the sum of $V_X, V_P, V_{PX}$ in Adlam 2022a (which is equivalent to $E_x V E[f(x) | P,X]$), and $V_{noise}$ is the sum of the rest (which is equivalent to $E_x E V[f(x)| P, X]$). In our scenario further decomposing $V_{clean}$ and $V_{noise}$ is not necessary since $V_{noise}$ already captures all the effects of label noise on test loss. In their Figure 1(j), Adlam et al plotted the seven terms $V_X, V_{\epsilon}, V_P, V_{X\epsilon}, V_{PX}, V_{P\epsilon}, V_{PX\epsilon}$ for the ridgeless setting ($\lambda=0$, Corollary 1) where the noise dependent term (sum of purple and green) doesn’t capture the ascent. And it still remains unclear how noise changes the monotonicity of the test loss curve when $\lambda \neq 0$.
>
> ### bias-variance decompositions for losses other than MSE
> We thank the reviewer for bringing up this paper. We clarify our sentence on page 6, as follows: “we chose MSE loss because its bias and variance can be empirically measured in an unbiased way”. Decomposition of functions in the Bregman divergence family is indeed well-defined, e.g, Yang’s equation (3) for CE loss. However, it still cannot be measured by an unbiased estimator. That’s the reason why we considered MSE loss to avoid systematic errors in the result.
>
> ### Reference
>
> Belkin, Mikhail, Daniel Hsu, and Ji Xu. "Two models of double descent for weak features." SIAM Journal on Mathematics of Data Science 2.4 (2020): 1167-1180.
>
> Hastie, Trevor, et al. "Surprises in high-dimensional ridgeless least squares interpolation." The Annals of Statistics 50.2 (2022): 949-986.

---

> ### Author Response · Authors · 2022-11-14
> **Response to Reviewer fFmQ**
>
> ### Clarification on our novelty and relation to previous results
> We respectfully disagree with the reviewer about the limited novelty of our work. Indeed, our work sheds light on several important and perhaps unexpected phenomena that have not been observed before. Given the recent focus in the community on the benefit of increasing model size, our work delivers an important message that increasing size can potentially hurt generalization, especially under label noise and small sample size.
>
> Below, we clarify our novel observations and theoretical analysis compared to prior work.
>
> **We are the first to**:
> - Empirically observe and theoretically formalize the *phenomenon of the final ascent*, i.e., larger model size harms the generalization in presence of label noise.
> - Incorporate model density into the theoretical analysis of double descent. We showed that while density and width can both control the model size, interestingly reducing density is more effective in improving robustness against label noise, i.e., the *optimal width at a smaller density achieves better generalization* (see Fig. 4(b, d) and 7).
> - Empirically observe that, perhaps surprisingly, *robust learning methods exacerbate the final ascent* instead of alleviating it, i.e., they make the final ascent happen with smaller model size.
> - Theoretically show the dependence of final ascent not only to the noise level but also to the *sample size* (Sec 3.1 and 5), and empirically confirm that increasing sample size eventually alleviates the final ascent.
>
> **Our theoretical novelty**:
> Theorem 1 cannot be obtained from taking the limit of previous work. The setting in our paper differs from previous ones that aim at deriving non-asymptotic results. We chose this setting as it allows incorporating the three important sources of randomness, namely training data, label noise, and random features that mimics the randomness in initialization and optimization (as opposed to e.g. models with fixed design or without random feature), and allows deriving interpretable closed form results (as opposed to non-linear feature or n/d=const limit). We are not aware of any non-asymptotic result that gives explicit expression of the bias-variance tradeoff, as we derived in this setting. We give a few examples that the final ascent has not been reflected in previous theoretical models on double descent:
> (Belkin et al. 2020): Corollary 2.2 (Gaussian model with random selected features) doesn’t capture the final ascent since the noise dependent term always decreases with the width. The `prescient’ selection model (their Fig 2) has an ascent in the end, however, as we plotted in our new Figure 26 in Appendix F, (1) the ascent exists even without noise, thus the result doesn’t show the double descent phenomenon in the first place, and (2) the ascent diminishes as we increase noise, and therefore shows an effect opposite to our empirical observation. The Fourier series model in Belkin’s Theorem 3.1 used to show double descent is for noise free setting.
> (Hastie et al. 2022): Among the models they considered, only the minimum-norm least squares with isotropic features can manifest a U-shaped curve in the overparameterization regime (their Fig 1). However, the effect of noise here is exactly the opposite of what we observed. As is described in their Sec 3.2, in the overparameterization regime, the loss monotonically decreases when SNR<1 (noise is large) and has a local minimum when SNR>=1 (noise is small).
>
> **Our analysis compared to Yang et al. 2020**: As discussed above, we chose the asymptotic settings of Yang et al. to be able to derive interpretable closed form results of the effect of label noise. Note that Yang et al. did not consider label noise in their analysis and therefore did not observe the final ascent. Importantly, our analysis incorporates the effect of density on the generalization (Theorem 2), *for the first time*. Originally the layer parameter is a vector and reducing width and reducing density are the same. To analyze the effect of density, we added an extra layer to the network so that the trainable layer has multiple output dimensions (considering layer parameters as a matrix, reducing width deletes rows while reducing density deletes individual entries). By analyzing the minimizer of the regression loss, we showed that the mask applied to the trainable parameters scales down the random features in the limit. Hence, reducing density effectively scales up the l2 regularization parameter (in Sec 4.3 we further discuss that reducing density may have an effect beyond regularization in neural networks). This makes our analysis distinct from the recent theoretical studies on overparameterization that mostly focuses on the effect of width.

---

> > ### Comment · Reviewer_fFmQ · 2022-11-18
> > **Response to authors**
> >
> > I'm grateful for the authors' time in responding to my comments. However, I do not feel that my concerns about the paper's novelty have been addressed.
> >
> > Specifically, I have verified that Theorem 1 of the paper can be derived directly from Theorem 1 of Adlam 2022a. Taking the two equations in (21) of Lemma 1 in Adlam 2022a, one can be used to eliminate tau_2 and get an equation for only tau_1 (call it EQ1). EQ1 can be differentiated with respect to gamma to get a second equation, EQ2. Next one can use the resultant of EQ1 and EQ2, to attempt to solve for V_noise. Finally setting gamma=gamma_0 / phi and taking the limits phi -> 0, zeta, eta -> 1 in resultant gives a quadratic equation for V_noise that has the same solution as in Theorem 1 of this paper. The algebra is messy but straightforward.
> >
> > While there are other contributions in the paper, I don't feel they are sufficient. Moreover, the discussion around Theorem 1 should be updated to reflect that it is a special case of previous work.

---

> > > ### Author Response · Authors · 2022-11-19
> > > **Response to Reviewer fFmQ**
> > >
> > > We thank the reviewer for pointing out the connection to Adlam 2020. However, we note that the final ascent phenomenon has not been revealed by Adlam 2020 since they only studied the shape of the loss curve for the ridgeless setting where the loss decreases in the overparameterization regime regardless of the noise. We have updated the main paper and included the discussion in Sec 3.1. We emphasize that the point of Theorem 1 is to provide a theoretical explanation for our main finding, i.e., the final ascent phenomenon. Whether it can be derived from Adlam 2020 or extended from Yang 2020 does not weaken our main contribution. As we mentioned in the earlier reply, **establishing the final ascent phenomenon which has been missed in the literature is indeed very novel and impactful. Moreover, our paper uncovers an richer understanding of the effect of**:
> > > - (1) **model density and width affect generalization differently: the optimal width at a smaller density achieves better generalization**
> > > - (2) **increasing sample size alleviates or remove the final ascent**
> > > - (3) **robust learning algorithms exacerbate the final ascent**
> > >
> > > All of the above are distinct from previous work. Therefore we hope the reviewer can reconsider the novelty of our paper as well as the score.

---

### Official Review · Reviewer_gtPU · 2022-10-26

**Confidence:** 3
**Correctness:** 4
**Technical Novelty And Significance:** 3
**Empirical Novelty And Significance:** 3
**Recommendation:** 8

**Clarity, Quality, Novelty And Reproducibility:**

**Clarity** The paper is very clear

**Quality** The paper is of good quality

**Novelty** The paper builds on existing framework but has new contribution

**Reproducibility** Seems good


**Strength And Weaknesses:**

### Strength
1. The proposed phenomenon is timely for the discussion of scaling and the message that larger model can hurt potentially generalization performance is very important
2. The theoretical component seems technically solid
3. The experimental results on deep neural network are thorough and match the prediction of the proposed random feature regression model
4. The observation that robust methods exacerbate final ascent is extremely interesting and could lead to fruitful future research
### Weakness
1. The phenomenon only becomes relevant with extremely large label noise (> 60%) which makes it less interesting for most practical cases where we can expect a reasonable signal-to-noise ratio.
2. The plots for 5c seem to contain fewer settings than the other ones. Why? I personally would be interested to see more
3. There is a gap between the regression setting of the analytical model and classification setting of the experiments.
4. It also seems like architectures can have a non-trivial effect on the phenomenon.


**Summary Of The Paper:**

The paper demonstrates the existence of a phenomenon called “final ascent” where noisy labels can hurt the generalization models with larger capacity, which contrasts with “double descent” where increasing the model capacity first hurts the model performance and but improves the performance again. First, the paper theoretically proves the existence of the phenomenon in the recently popular random feature regression framework. Roughly speaking, the result shows that in the presence of label noise, the risk of the analytical solution has an extra term governed by the noise-to-sample size ratio. With similar technique, the paper also proves a theorem that shows sparsity can alleviate final ascent. In the second part, the paper conducts a series of experiments on MNIST, CIFAR10/100 and MLP, LeNet and ResNet34 and show that realistic models and data also exhibit final ascent.

**Summary Of The Review:**

The paper makes a solid theoretical contribution to the line of work on random feature models and conducts thorough experiments. I do not have major complaints about the paper and thus recommend accept.

---

> ### Author Response · Authors · 2022-11-14
> **Response to Reviewer gtPU**
>
> We thank the reviewer for acknowledging our contribution.
>
> ### Noise ratio required for the final ascent to occur
> We emphasize that while a relatively high **symmetric label noise** is required to observe the final ascent, a **moderate noise** level results in the final ascent under the following *more realistic* scenarios: (1) when noise is **asymmetric** (Fig 5, Fig 9 (a) and  Fig 16), or when (2) **robust methods** are applied (Fig 9). We expect the phenomenon to be even more pronounced under stronger types of noise, i.e., manual-labeling mistakes (usually class/instance-dependent and more skewed than symmetric label flipping) or adversarial noise.
>
> ### The plots for 5c
> In Figure 5(c) we plotted the loss under 40% asymmetric noise. Since the phenomenon was already obvious and provided enough evidence for our main finding we didn’t think it’s necessary to conduct more experiments. Now, to further confirm that under asymmetric noise, increasing noise does exacerbate the ascent, we present new experiments on CIFAR-10 under 30% asymmetric noise in Figure 16 (c)(d). We see that the loss increases only slightly from width 128 to 256. As expected, the final ascent here is less pronounced compared to that under 40% noise in Figure 5(c).
>
> ### Regression vs classification
> To characterize the final ascent in modern machine learning, the analysis must successfully capture double descent without noise in the first place. The double descent phenomenon has only been *theoretically characterized* in regression tasks and only empirically observed in classification tasks. For this reason we took ridge regression for our theoretical analysis. However, as we show in the paper, our theory can indeed give us useful insights and match the empirical observations in classification tasks (e.g., Fig 3 shows that our theory can explain the shape of the variance curve). We thank the reviewer for bringing this up, and agree that theoretically characterizing double descent and other related phenomena in classification problems can be an interesting direction for future work.
>
> ### Effect of model architecture
> This is a great observation. Model architecture can indeed significantly affect the generalization performance under noise, since it provides certain inductive bias when combined with learning algorithms. Recently, [1] has discussed how architecture affects the quality of learned representations under label noise. And we also notice that a contemporary submission https://openreview.net/forum?id=7i6OZa7oij has shown that inductive bias of a model can determine whether interpolation of noisy data is harmful or not. However, our work is orthogonal to the studies on model architecture. In particular, we showed that the final ascent can be observed in **any architecture**, by increasing its size. We believe that it would be an important future direction to combine these two lines of research and precisely characterize how model architectures affect the test loss vs. model size.
>
> [1] Li, Jingling, et al. "How does a Neural Network's Architecture Impact its Robustness to Noisy Labels?." Advances in Neural Information Processing Systems 34 (2021): 9788-9803.

---

### Official Review · Reviewer_romc · 2022-10-27

**Confidence:** 5
**Correctness:** 3
**Technical Novelty And Significance:** 3
**Empirical Novelty And Significance:** 3
**Recommendation:** 6

**Clarity, Quality, Novelty And Reproducibility:**

(Clarity) This work is well presented.

(Quality) High quality work, makes important contributions.

(Novelty) Novel.

(Reproducibility) Good.

**Details Of Ethics Concerns:**

N/A.

**Strength And Weaknesses:**

**Strength**:

1. This paper identifies two interesting findings on the effect of “overparameterization under label noise:

>(1). the variance of the generalization loss has a second ascent when the noise is large; and

>(2). the robust training methods (robust towards label noise) make the last ascent more pronounced.

These two findings could be practically useful for handling label noise, especially when we apply robust training methods for training highly overparameterized models.


**Weaknesses**:

1. [minor] There seems to be only one point for demonstrating the 'last descent' phenomenon on CIFAR10 (as shown in Figure 2 and Figure 5(b)). It would be interesting to study an even wider network to further examine the trend (if computation allows).

**Summary Of The Paper:**

This paper studies the role of overparameterization for generalization when training with noisy labels. Specifically, the authors focus on the noisy label training data setting and investigate how test loss of model changes with respect to different levels of overparameterization. This paper provides two interesting empirical findings: (1). 'the variance of the generalization loss experiences a second ascent when the noise-to-sample size ratio is large'; (2). when applying robust training methods (robust towards label noise), the robust training methods make the last ascent more pronounced. The paper also provide theoretical analysis for the first finding in a simplified setting.

**Summary Of The Review:**

This paper provide interesting theoretical and empirical findings on overparameterization under label noise, I would recommend acceptance.

---

> ### Author Response · Authors · 2022-11-14
> **Response to Reviewer romc**
>
> We thank the reviewer for acknowledging the novelty and importance of our work. To make the trend more clear, we ran width 192 and 384 (the largest one that can be trained within a reasonable time) on CIFAR-10 with 80% noise and updated Fig 5(b) and 2. We see that both the loss and variance increase from width 192 to 384.

---

> > ### Comment · Reviewer_romc · 2022-11-30
> > **Thanks for the response**
> >
> > Thanks for the response. The updated experiments addressed my concern.

---

### Decision · Program_Chairs · 2023-01-20

**Decision:**

Reject

**Justification For Why Not Higher Score:**

The negatives raised by the reviewers were on how novel the theoretical results were. Two of the reviewers note that the theorems follow from existing work. While I don't find this novelty a gigantic issue, it is one that requires careful writing. The second point raised by the reviewers is a gap between theory, done with additive Gaussian noise, and experiments that focus on classification noise. The authors point out that this limitation is not special to their work and is true for other characterizations of double descent. One question is why not study regression? In the recommendation for reject is that there are lots of clarity and framing problems that will take effort to clean up. As another example, the results on robust algorithms in section 6 were quite interesting, but how they fit into the rest of the paper was less clear.

**Justification For Why Not Lower Score:**

N/A

**Metareview: Summary, Strengths And Weaknesses:**

The main finding here is to study how bigger models can generalize worse when there is noise in the labels. At a high,level a-priori, this finding is not surprising because larger models are more likely to interpolate, which means they are more likely to fit noise. The added angle is a focus on the size of the overparameterized model, where there are several results showing improved generalization. This question is studied theoretically in the random feature regression model with a generalization decomposition that yields an increasing noise dependent term. There's an empirical study considering width on both CIFAR and MNIST.

Reviewers of this paper had expertise in the area, but ended up with differing options ranging from accept to reject. The positives were that the empirical demonstration of this phenomenon was important and a simplified technical analysis. The negatives raised by the reviewers were on how novel the theoretical results were. Two of the reviewers note that the theorems follow from existing work. While I don't find this novelty a gigantic issue, it is one that requires careful writing. The second point raised by the reviewers is a gap between theory, done with additive Gaussian noise, and experiments that focus on classification noise. The authors point out that this limitation is not special to their work and is true for other characterizations of double descent. One question is why not study regression? In the recommendation for reject is that there are lots of clarity and framing problems that will take effort to clean up. As another example, the results on robust algorithms in section 6 were quite interesting, but how they fit into the rest of the paper was less clear.